# Establishing flood thresholds for sea level rise impact communication

Sadaf Mahmoudi ®[1,2] ✉, Hamed Moftakhari ®[1,2] ✉, David F. Muñoz[3], William Sweet[4] & Hamid Moradkhani ®[1,2]

Sea level rise (SLR) affects coastal flood regimes and poses serious challenges to flood risk management, particularly on ungauged coasts. To address the challenge of monitoring SLR at local scales, we propose a high tide flood (HTF) thresholding system that leverages machine learning (ML) techniques to estimate SLR and HTF thresholds at a relatively fine spatial resolution (10 km) along the United States' coastlines. The proposed system, complementing conventional linear- and point-based estimations of HTF thresholds and SLR rates, can estimate these values at ungauged stretches of the coast. Trained and validated against National Oceanic and Atmospheric Administration (NOAA) gauge data, our system demonstrates promising skills with an average Kling-Gupta Efficiency (KGE) of 0.77. The results can raise community awareness about SLR impacts by documenting the chronic signal of HTF and providing useful information for adaptation planning. The findings encourage further application of ML in achieving spatially distributed thresholds.

High tide flooding (HTF, a.k.a. sunny day flooding or nuisance flooding) is a category of minor floods driven by elevated coastal sea levels with a potential for business and traffic interruptions and infrastructure degradation, without considerable structural damages or causalities[1]. At the incident level, HTF may not pose great damage to the coastal communities. However, monitoring its evolution over time and assessing its associated impacts on the resilience of coastal communities is crucial, as over the long run it can aggregate to cause structural instability issues and damage, among other chronic impacts. The cumulative costs associated with HTF pose serious policy challenges that require comprehensive monitoring and communication of the chronic impacts of HTF on socioeconomic well-being, infrastructure resilience, and public health[2,3]. Previous studies examining still water levels recorded by tide gauges of the National Oceanic and Atmospheric Administration (NOAA) have revealed a discernible increase in the occurrence and duration of HTF events at monitored sites along the coastal regions of the United States[4–6]. Sea level rise (SLR), as a result of climate change[7–9], is known as the primary factor behind the increased frequency of HTF[4,5,10–15], which contributes to adverse consequences in low-lying coastal regions[12,16,17].

As SLR persists, even if the Paris Agreement targets are achieved, it is imperative for coastal communities and the government to implement adaptation strategies in order to combat the increasing occurrence of HTF in the U.S.[7,12] However, creating adaptive strategies to mitigate SLR and HTF is complex and it requires a solid understanding of SLR's evolving patterns and their associated impacts at local and regional scales[18]. The local estimates of relative SLR would deviate from the estimated accelerating global rates ($3.1 \pm 0.3$ mm/yr[16,19–21]) due to local/regional processes[22]. To facilitate effective long-term planning, it is imperative to prioritize the development of local projections for sea level analysis and prediction[18].

Considerable progress has been made in projecting SLR using hybrid approaches that combine process-based modeling with statistical methods. These approaches enable us to quantify uncertainties and likelihoods of various SLR projections under different climate change scenarios. However, there is a need to communicate the severity of predicted impacts more effectively. There is a general absence of reliable and generalizable metrics that allow for ongoing monitoring of SLR impacts across spatiotemporal scales. The availability of such metrics would aid in quantifying SLR impacts and provide those responsible for

[1]Center for Complex Hydrosystems Research, The University of Alabama, Tuscaloosa, AL, USA. [2]Department of Civil, Construction, and Environmental Engineering, The University of Alabama, Tuscaloosa, AL, USA. [3]Department of Civil and Environmental Engineering, Virginia Tech, Blacksburg, VA, USA. [4]NOAA/National Ocean Service, Silver Spring, MD, USA. ✉e-mail: smahmoudikouhi@crimson.ua.edu; hmoftakhari@ua.edu

communicating the effects of climate change with tangible metrics that are more easily understood and relatable to the public.

Our current understanding of SLR rates and HTF thresholds is heavily reliant on measurements from point-based gauges. In order to establish HTF thresholds, which represent the coastal water level at which nuisance flooding impacts begin to occur for coastal communities, a systematic approach is currently adopted. This involves the utilization of two key components: local tidal records and a local flood monitoring system. Firstly, local tidal records serve as a valuable source of information for determining the baseline water levels in a coastal area. These records provide insights into the regular tidal patterns and can help establish a starting point for measuring HTF thresholds. Satellite altimetry data complement in-situ tide gauge observations by providing a long-term still water level record at a relatively fine temporal resolution (i.e., hourly) and offer the most comprehensive observational record of historical sea level trends[21,23–25]. However, relying solely on tidal records may not be sufficient to monitor evolving patterns. To accurately identify when nuisance flooding impacts begin to occur (e.g., road and business closures), it is imperative to implement a local flood monitoring system. This system is designed to continuously track and record favorable conditions at which flood impacts occur. By analyzing data collected from the flood monitoring system, coastal communities can identify specific water levels at which nuisance flooding is observed. These observed conditions serve as critical reference points for establishing HTF thresholds[26,27].

Nevertheless, some challenges are inherent in the aforementioned common approach. For example, tide gauges are often few in number and unevenly distributed in worldwide coastlines. Along the U.S. coasts, there exist thousands of communities without a tide gauge in close proximity. If the US coastlines were segmented into 10 km intervals, accommodating variations in coastal characteristics, 75% of the coastal communities in the United States lack access to tide gauges within a 10 km radius of their respective locations (Supplementary Fig. S1). This observation underscores the limited coverage of and accessibility to tide gauge infrastructure along the U.S. coastlines, thereby hinders emergency managers and stakeholders from accessing reliable local information including HTF thresholds and SLR rates. Consequently, They make assumptions and use the information available at the nearest tide gauge that might be located more than a hundred kilometers away, with its own accuracy and uncertainty challenges[28]. The utilization of nearest tide gauge values necessitates careful consideration, as it might overlook the underlying patterns of the influential features on the HTF thresholding system. This approach presumes that the target ungauged point and the nearest official threshold location inherently share consistent characteristics in terms of the influential variables. This presumption, however, does not hold true across a substantial stretch of the coast, where effective variables might pose divergent conditions on HTF thresholding system. Also, it is worth noting that the lack of a comprehensive flood monitoring system over the past few decades in many tide gauges does not allow for a systematic delineation between flooding and non-flooding conditions that provide crucial information for HTF thresholding.

Sweet et al.[5] identified a lack of uniformity in representing flood impact severity, even in areas with established thresholds, and introduced a normalization method to universally categorize and assess different flood events, ensuring consistency. They proposed a linear relationship between thresholds above mean lower low water (MLLW; the average of the lower low water height of each tidal day observed over the National Tidal Datum Epoch) and greater tidal ranges (GTR; which is the difference between tides above MLLW and MHHW datum; $GTR = HTF_{MLLW} - HTF_{MHHW}$)[29,30] as a means to establish a globally applicable HTF thresholding system:

$$HTF_{MLLW} = 1.04 \times GTR + 0.5 \quad (1)$$

Their method measures the HTF above MLLW, mainly because NOAA's Weather Forecast Office (WFO) originally sets similar thresholds based on MLLW. However, proposing the HTF threshold above MLLW (i.e. $HTF_{MLLW}$) would have an embedded GTR that might overshadow the variability of the actual flooding threshold above the mean higher high water (MHHW; the average of the higher high water height of each tidal day observed over the National Tidal Datum Epoch) in places with larger tidal ranges. MHHW should be the preferred and more logical vertical datum of reference when analyzing floods, because locally, MHHW works as a proxy for the high tide that coastal communities expect the coastal water level to reach on a regular basis. Thus, a better formulation of HTF threshold above local high tide datum ($HTF_{MHHW}$) is:

$$HTF_{MHHW} = HTF_{MLLW} - GTR = 0.04 \times GTR + 0.5 \quad (2)$$

Figure 1a shows the results of the linear regressions above MLLW and MHHW datums between the officially reported HTF thresholds at 70 NOAA tide gauges along the U.S. coasts with their associated GTR. The magenta and purple dots and the associated dashed lines show officially reported thresholds by NOAA and the linear regression results, above MLLW and MHHW datums, respectively, along with their goodness of fit metrics reported inside the box of the same color. The difference between estimates based on Eqs. (1) and (2) relies on the different datum above which the threshold is measured. Indeed, an insufficient skill of Eq. (2) to estimate $HTF_{MHHW}$ is evident (i.e. with a negative NSE), especially for points with larger GTR. Moreover, this means a linear regression solely based on one independent variable (here GTR) generally fails to capture the stochastic nature of spatial variability in HTF threshold above the local high tide datum (i.e., MHHW). This approach, if generalized as done in previous studies[12,13,28,31,32], could yield in a large error in estimated flood frequency. Therefore, the variability in HTF thresholds necessitates the incorporation of multiple features, and a simple linear regression method proves insufficient to unravel the complex patterns inherent in these thresholds.

These facts motivated us to develop an approach that provides flood thresholds above the MHHW datum (i.e., $HTF_{MHHW}$) based on multiple physically relevant variables, which can inform decision-makers and emergency managers about the patterns and efficiency of mitigation plans coping with nuisance flooding. The machine learning (ML) algorithms developed for this purpose provide spatially distributed information on the height of $HTF_{MHHW}$ that is essential for tracking patterns and trends of HTF over space. ML algorithms can effectively process and analyze numerous input features simultaneously. In the last two decades, the use of ML methods for extracting characteristics, trends, or rules from datasets has been rapidly increasing for data-intensive hydrological modeling problems[33,34]. Moreover, the observed variability in the components that impact the HTF thresholding system has made it necessary to apply clustering techniques. These techniques help partition effective variables into distinct categories classified by shared characteristics. The clustering approach has played a pivotal role in generating more accurate and refined outcomes pertinent to HTF thresholds. The Random Forest (RF) regression model was selected as an appropriate ML model because of its ability to identify underlying patterns and predict the values of HTF thresholds at ungauged basins. One of the influential variables needed to estimate the HTF threshold at high spatial resolution is the local SLR rate. This rate includes essential information on the time evolution of coastal sea levels and local processes that influence long-term variabilities in the HTF threshold. Recent efforts have been made to achieve a high spatial resolution in SLR estimates with the help of ML algorithms[35,36].

Here, we show the continental-scale estimates of high spatial resolution data on relative SLR rates and HTF thresholds, which are

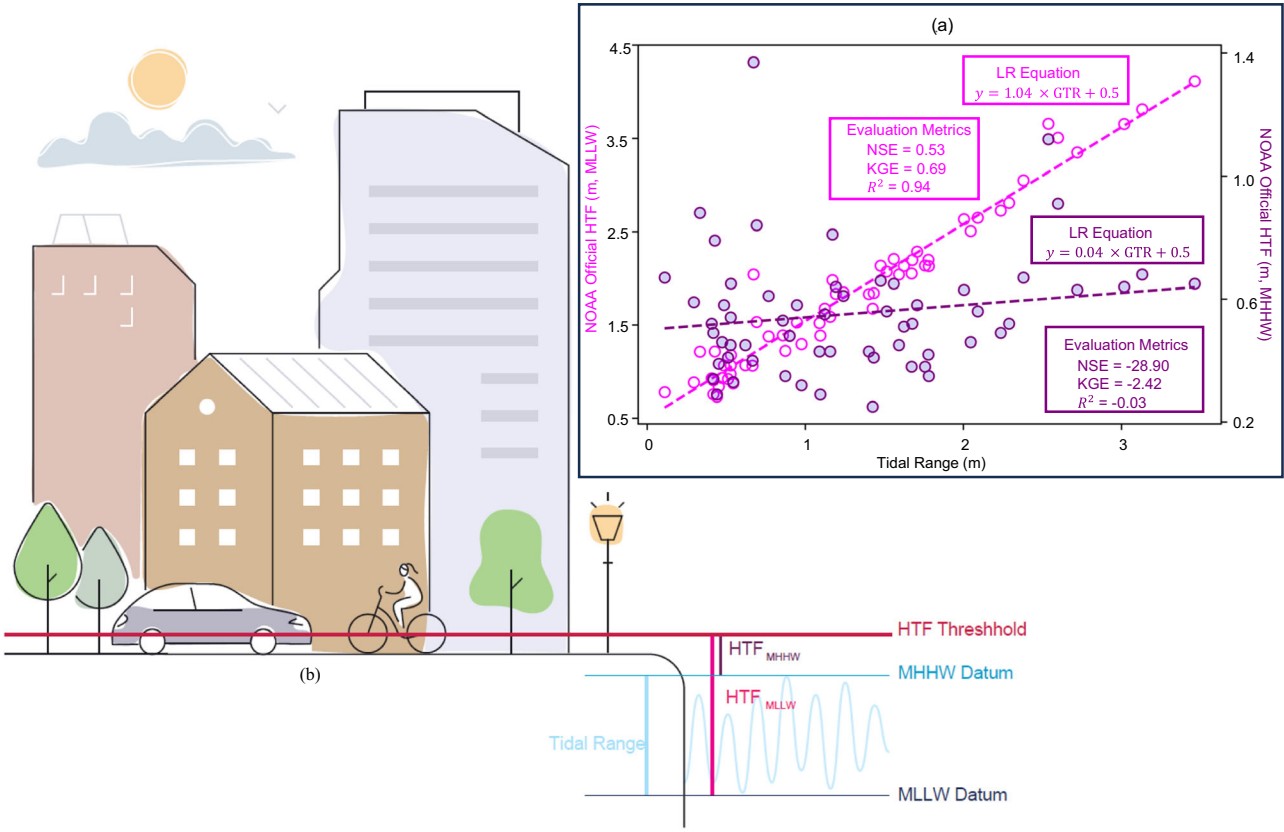

**Fig. 1 | HTF threshold estimation. a** Comparison between the relationships of HTF thresholds with GTR when measured above MLLW vs. above MHHW; Magenta dots and the left y-axis show $HTF_{MLLW}$; purple dots and the right y-axis represent $HTF_{MHHW}$; **b** tidal characteristics and datum over a typical tidal cycle (Sadaf Mahmoudi - stock.adobe.com).

intended to facilitate the communication of SLR risk to vulnerable coastal communities, particularly in locations that lack gauge measurements. Our findings, utilizing a well-trained and validated ML algorithm, provide reliable information at a spatial resolution (10 km along the U.S. coastline) considerably finer than the previously available datasets[5,12]. This information plays a pivotal role in effective risk-informed decision-making and communication of trending flood hazards associated with SLR to at-risk coastal communities.

## Results

### Clustering

We developed and trained multiple ML algorithms to learn and predict SLR rates and HTF threshold patterns along the coasts of the United States. To provide spatially variable coastal dynamics characteristics, we used the K-means algorithm[37] to cluster the HTF threshold's input data into different categories. The elbow approach[38] is used to determine the optimal number of clusters. This approach plots the number of clusters against the total distance between data points and their respective cluster centroids and identifies the elbow point on the plot, which indicates a trade-off between minimizing the distance and mitigating the complexity associated with an excessive number of clusters. Based on the results of the elbow approach (Fig. 2a), three different clusters were required for the dataset to be divided on. Figure 2b shows the geographical spread of clusters over the U.S. coastline. The K-means algorithm clustered the study locations into the following three regions based on regional similarities of input features: i) West Coast, ii) Gulf and Southeast Coasts, and iii) Northeast Coast. This is consistent with physical features in these systems that govern the still water level dynamics and the associated HTF[6].

### SLR algorithm

We first trained and validated the ML model against data at existing gauges to further estimate the rates of SLR at ungauged coasts. Feature transformation and randomized search[39,40] are essential steps in the process of ML algorithm development. The tuned hyperparameters for each region, climate scenario, and percentile are presented in Supplementary Table S2 (see Supplementary Files). Additionally, RF regression algorithm[41] was used to estimate the SLR rate at 10 km intervals along the U.S. coasts. To test the developed RF regressor, the predicted SLR rates were compared to the target values from Kopp et al.[22]. Supplementary Table S3 presents the prediction accuracy of the algorithms based on different performance metrics. Figure 3a illustrates a comparison between the observed rates of SLR and the rates predicted by the trained algorithm. This comparison supports the suitability of the algorithm for estimating the spatially distributed rates of SLR.

Furthermore, the box plots in Fig. 3b summarize the observed vs. estimated rates of SLR at the three study regions/clusters under the current climate. The suitability of the proposed ML algorithm for the estimation of SLR is further supported by the regional summary statistics presented. These statistics reveal interesting patterns that are in line with previous studies[12]. For instance, the Northeast cluster shows the largest regional median rate of SLR, which is 5.25 mm/yr, as well as the smallest standard deviation when compared to the other two studied clusters. In contrast, while the regional median of relative SLR rates in the Gulf and Southeast cluster is not much smaller than the one in the Northeast cluster, the distribution is negatively skewed and substantially wider. This indicates that the cluster encompasses sites with estimated SLR that are remarkably greater than the median value. Similarly, the West Coast cluster shows a smaller median value of

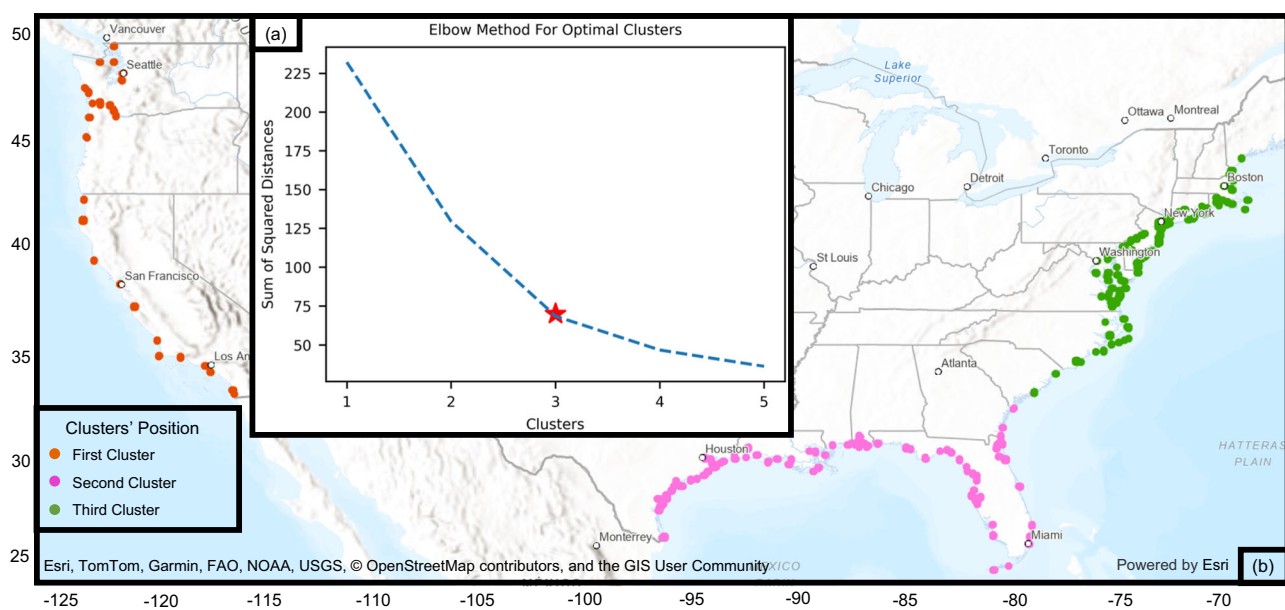

**Fig. 2 | K-means clustering. a** Optimal number of clusters for the U.S. coastlines using the elbow approach; **b** Cluster locations: West Coast, Gulf and Southeast Coasts, and Northeast Coast.

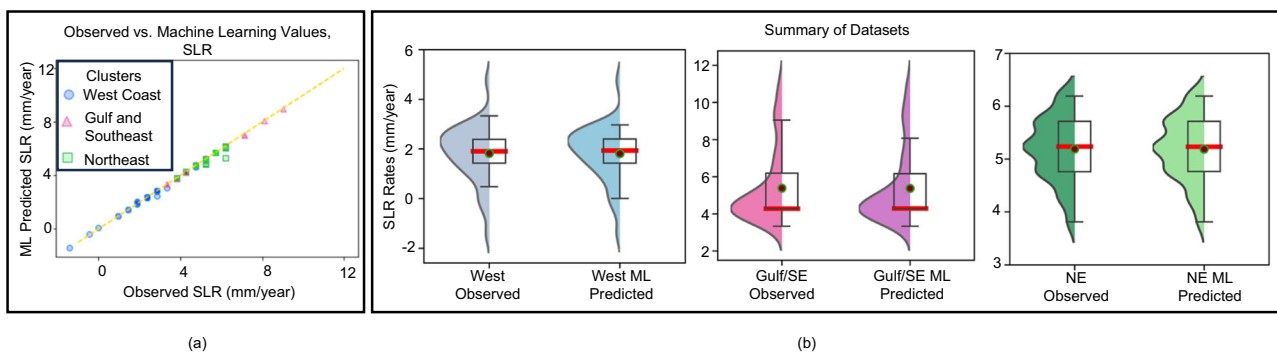

**Fig. 3 | Validation of ML algorithms for HTF threshold estimation. a** Comparison between the observed and predicted SLR rates for the current conditions; **b** The statistical summary between the observed and predicted SLR rates in the West Coast, Gulf and Southeast Coasts, and Northeast Coast. The circles, triangles, and square symbols in **a** show the rates related to the West coast, Gulf and Southeast coasts, and the Northeast coast, respectively, and the box plots in **b** show the median, quartiles, maximum, and minimum of each dataset. The violin plot on the left of each box plot shows the Kernel Density plot.

2 mm/yr and a relatively wide positively skewed distribution. This distribution includes locations with rates of relative SLR that are below the global average rate of SLR. Taken together, these regionally consistent patterns further support the applicability of the proposed ML algorithm for the estimation of spatially distributed rates of SLR when trained regionally.

To further investigate the impact of different input features on SLR rates, we conducted an analysis of their relative importance under each scenario, as shown in Supplementary Table S4. Based on these results, the West Coast is most affected by the latitude; the Gulf and Southeast Coasts are impacted by the ocean circulation the most; and the most important feature in the Northeast Coast is latitude and vertical land motion (as suggested by Ohenhen et al.[42]). To determine the importance of the contribution of each feature, we eliminated the variables with the least importance from the feature dataset and then retrained the model. However, we found that the evaluation metrics worsened, indicating that all variables were essential as input data. This underscores the fact that even features with relatively low importance at the regional scale can still exert prominent local influences. Our analysis showed that vertical land motion (VLM) had the least importance in affecting SLR rates in the Gulf and Southeast cluster. This

finding is in contrast to previous studies, such as Jankowski et al.[43], which have conducted local analyses of the drivers of SLR. One possible explanation for this discrepancy is that there is a lack of observational water level data in some parts of the Gulf Coast, such as Louisiana, where considerable vertical land motion is occurring. Only three tide gauges with long enough records exist along the coast of Louisiana that enable the calculation of relative SLR rates. As a result, our ML algorithm may not have sufficient information to learn the importance of VLM in this region, highlighting the need for more localized and detailed data to improve the accuracy of data-driven models like the one proposed here.

## HTF algorithm

We went through an identical training and validation procedure to ensure the suitability of the proposed ML algorithms for estimating spatially distributed HTF thresholds. Supplementary Table S5 shows the tuned hyperparameters under each scenario. Figures 4a and b show the comparison between the observed and predicted values using two different approaches: the ML algorithm presented in this study and the linear regression (LR) introduced by Sweet et al.[5] (Eq. (2)), respectively. Sweet et al.[5] thresholds have offered a narrower

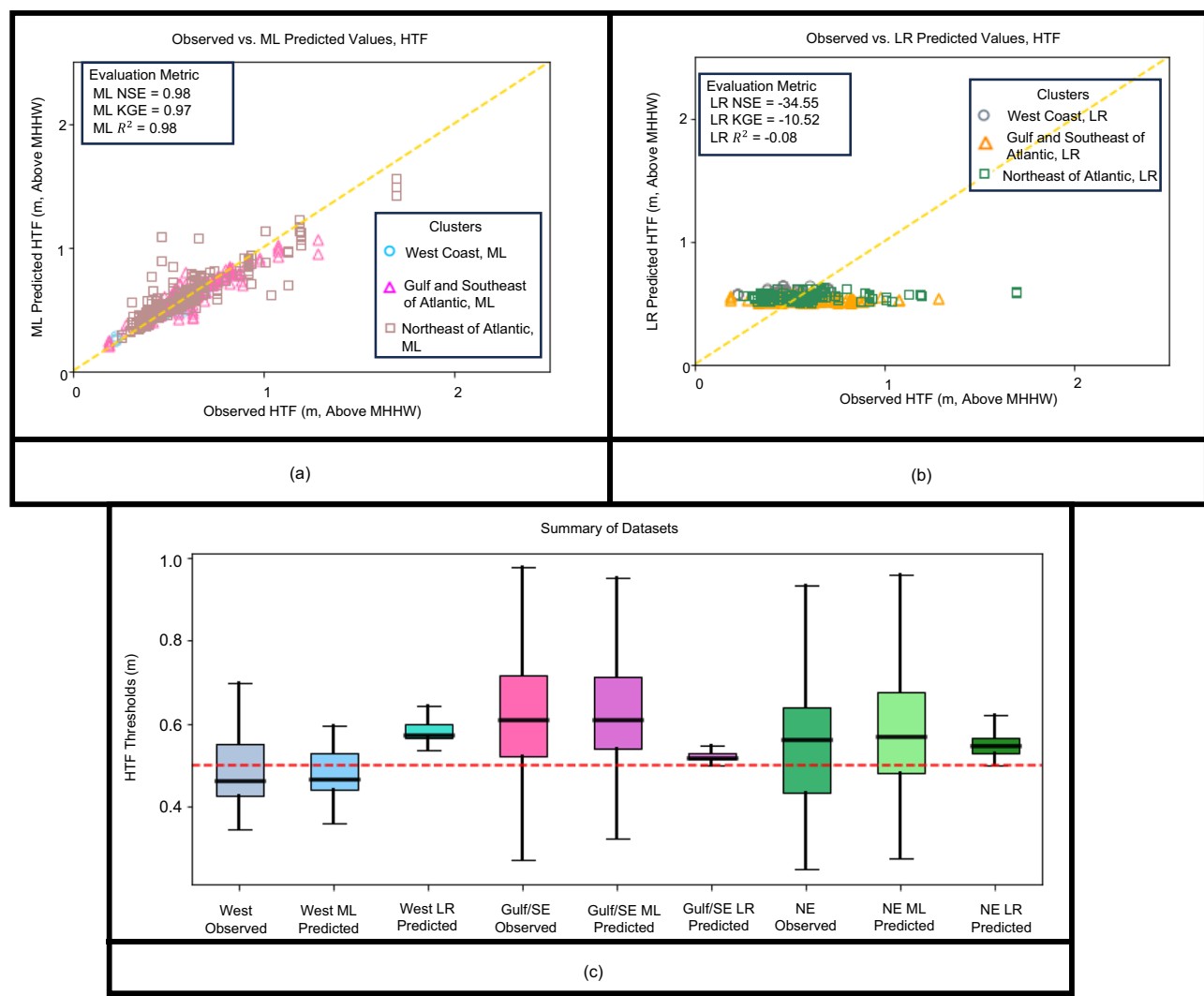

**Fig. 4 | Comparison between the observed HTF thresholds above MHHW and the two alternative estimation approaches. a** Machine learning (ML)- and **b** linear regression (LR)-driven HTF thresholds, **c** Boxplot for regional statistical summaries of observed/estimated HTF thresholds. The red dashed line in panel **c** represents the 0.5 m offset suggested by Sweet et al.[5].

range of the HTF threshold, when compared with the reported ones from the National Weather Service (NWS). It is partially due to the normalization scheme used in their regression-based methodology, and the fact that they do not aim perfectly mimic the reported NWS thresholds. The proposed ML algorithm here, on the other hand, puts more trust in the correctness of the reported NWS thresholds and so is less constrained. The box plot of HTF thresholds (Fig. 4c) compares the statistical summary of observed, ML-predicted, and LR-predicted threshold values in the three studied regional clusters. The LR approach for estimating median and range values of HTF thresholds shows poor performance across various regions, with only the Northeast coast demonstrating a close match to reported median values. However, the LR approach fails to produce a percentile range that aligns with the reported values across all regions. In contrast, based on regional summary statistics, the ML algorithm performs relatively well in predicting both the median and the range of HTF thresholds. All the clusters show a symmetrical distribution, whereas the Northeast coast and the Gulf and Southeast coastal regions demonstrate a wider range of values. This regional variability of HTF thresholds highlights the limitations of current methods that rely solely on univariate linear regression[12,13,28,31,32] and calls for a shift towards more advanced nonlinear approaches. ML algorithms can

provide better estimates of HTF thresholds and help identify regional and continental patterns of HTF, which is crucial for effective coastal flood risk communication, adaptation planning, and management.

The feature importance obtained from the HTF threshold algorithms is reported in Supplementary Table S7. On the West Coast, latitude is the most important feature, given the region's elongated shape from south to north. In contrast, on the Gulf and Southeast Coasts, longitude has the most substantial impact on HTF thresholds, as the region is elongated from west to east; also, the second most effective feature is the latitude, indicating the importance of the geographical location. For the Northeast coast, the latitude is the most important feature, followed by GTR, which is greatly variable from 5.8 m at the northern coast of Maine to 1.7 m down in Charleston, South Carolina. The longitude, SLR, and GTR have close importance on both the West and Northeast coasts. During this stage of analysis, a necessary refinement was performed by removing two components, including coastal elevation and continental shelf slope, which were found to have relatively minimal impacts on the thresholding system. This strategic elimination of less influential factors led to improvements in the model performance and a decrease of computational time to train the RF regressor. As depicted in Supplementary Fig. S3, a compelling representation of the percentile ranking was developed

based on the disparities between the officially established thresholds by NOAA and the values predicted through the RF algorithms. In this context, disparities falling within the narrow range of ±0.05 m were assumed to be negligible. However, disparities exceeding the defined threshold were regarded as substantial deviations warranting attention. Supplementary Fig. S3 demonstrates that the percentile ranking associated with large disparities shows an increase when the coastal elevation and continental shelf slope were omitted from the input features. The bullet points in this figure show the difference between the NOAA official thresholds and ML-predicted target values. Supplementary Tables S6a and S6b show the evaluation metrics with and without considering the coastal elevation and continental shelf slope, demonstrating the improvement after removing these two components.

The performance metrics used in the validation of SLR estimation algorithms in the West coast, the Gulf and Southeast coasts, and the Northeast coast show a Nash-Sutcliffe efficiency (NSE) of 0.74, 0.95, 0.8, Kling-Gupta efficiency (KGE) of 0.82, 0.94, 0.85, mean absolute error (MAE) of 0.54, 0.32, 0.29, and R-squared of 0.77, 0.95, 0.85, respectively, under the current climate scenario. Regarding the HTF thresholds, the average performance metrics for the prevailing conditions in the West coast, the Gulf and Southeast coasts, and the Northeast coast were found to be NSE of 0.42, 0.4, 0.3, KGE of 0.6, 0.54, 0.48, MAE of 0.04, 0.08, 0.13, and R-squared of 0.52, 0.44, 0.36, respectively. The evaluation metrics for the validation step for the SLR rates under different climate scenarios are shown in Supplementary Table S8a. Supplementary Tables S8b and S8c show the average evaluation metrics in the validation process for HTF thresholds' algorithms with and without considering coastal elevation and continental shelf slope, respectively. This observation serves as evidence that the ML algorithm performed satisfactorily when presented with previously unseen samples and further supports the proposition of developing a comprehensive methodology to generate high-resolution SLR rates and HTF thresholds.

### Spatially distributed SLR rates and HTF thresholds

The overarching goal of this study is to provide spatially distributed information on SLR rates and HTF thresholds every 10 km along the Continental United States (CONUS) coastline. For this purpose, the trained and validated ML algorithms are used to generate the maps of the spatially distributed estimates of SLR rates and HTF thresholds (Fig. 5a, b, respectively). Moreover, the NOAA official thresholds are presented in Fig. 5b in larger bullets to facilitate the comparison between the spatially distributed data and the official thresholds. According to these spatially distributed estimates of SLR rates, the coasts of Louisiana and east of Texas have experienced higher rates of SLR, compared with the rest of the Gulf and Southeast coasts that have experienced milder rates of SLR closer to the global average rates. This finding aligns well with the previously reported rates[12,44], which showed that Texas and Louisiana experience relative SLR rates that are 2–4 times greater than the global average rates that could have been driven by various factors such as shallow sediment compaction, altered hydrology, and fluid extraction[44–46]. This is also observed in the other clusters, including the Northeast coast, with a median SLR rate of 5.4 mm/yr. The Pacific Northwest region displayed the lowest SLR rates among all other assessed regions, partially due to isostatic adjustments[47]. Overall, the general agreement of the summary statistics (Fig. 5a) with the reported regional rates in the literature[12] further supports the applicability of the proposed ML-based approach in providing reliable estimates of spatially distributed rates of SLR.

### Discussion

This study aimed at improving the existing coastal HTF thresholding system. The two commonly used approaches are 1) based on specific impacts at a given location, and 2) considering influential features on the thresholding system. While the former suffers from the consistency of severity of impacts between various locations, the latter approach aims at establishing a unified method applicable across all regions, therefore mitigating the necessity of establishing local monitoring stations to record flood impacts for threshold determination. Transitioning and enhancing the thresholding system from an impact-based local approach to a regionally feature-based methodology is imperative in advancing flood monitoring and management practices. To pursue this approach, Sweet et al.[5] developed a linear regression based on one variable, which was found to be constrained, and non-generalizable, especially in regions with higher GTR values, such as the Northeast coast of the United States. Moreover, focusing on a single variable in the HTF thresholding system may lead to negligence of other crucial factors.

The proposed methodology in this study represents a substantial advancement towards a global strategy that leverages ML algorithms to provide spatially distributed information efficiently. Among the influential components of the HTF thresholding system, SLR rates played a pivotal role, given their remarkable impact on coastal flooding events. However, the localized availability of SLR rates presented a challenge, demanding a deeper investigation into this critical variable. To address this challenge and derive local SLR rates and HTF thresholds, we employed a ML algorithm, which offers the advantage of including multiple effective variables.

The literature[12,44] suggested that SLR rates in Louisiana and Texas are 2–4 times higher than global average (3.1 ± 0.3 mm/yr[16][19–21]), which confirms our findings (Fig. 5b). These elevated rates are attributed to multiple factors, including shallow sediment compaction, hydrological changes, and fluid extraction[44–46]. On the other hand, SLR rates are lower than the global average rates on the West Coast, especially because of isostatic adjustments[47].

The outcome of this study differs from the results of the linear regressions (LR) method introduced by Sweet et al.[5], who attempted to normalize and constrain the NOAA official thresholds in a range. The LR method demonstrates a relatively acceptable performance primarily in areas where the HTF threshold hovers within approximately 0.5 meters above the MHHW. Nevertheless, it is noteworthy that our ML algorithm exhibits a capacity to generate reasonably accurate results across a more extensive spectrum of observed HTF threshold ranges. The natural differences in HTF thresholds across regions highlight limitations in current methods that mainly use univariate linear regression and suggest a need to shift to nonlinear approaches.

Based on Supplementary Table S7, one prominent factor affecting HTF thresholds in the Northeast coast is the GTR. Specifically, it is observed that GTR values are higher in the states of Massachusetts, New Hampshire, and Maine, which contributes to the elevated HTF thresholds observed therein. Moreover, based on Supplementary Table S7, SLR stands as a more influential component on the HTF thresholds than GTR in the Gulf and Southeast coasts. Hence, higher rates of SLR could generally lead to higher HTF thresholds, such as the east Texas, that exhibits high SLR rates and HTF thresholds. While Louisiana faces high SLR rates, the lack of comprehensive flood mitigation strategies has led to lower HTF thresholds in this region, partially owed to the impact-based thresholding system set by NOAA. This complex situation makes Louisiana more prone to HTF effects and underscores how the complex interplay of environmental factors and human actions shapes coastal vulnerabilities.

Some disparities are evident between HTF estimates in specific localities and their adjacent parts (Fig. 5b), which results in spatial heterogeneity near flood prone metros. For example, a flood protection provided by hurricane seawalls in regions from Corpus Christi to Galveston Bay in Texas yields in higher HTF thresholds in this region, compared with other parts of Texas and the coast of Louisiana. Furthermore, the flood mitigation strategies employed in New York made the HTF thresholds be higher than their surroundings. These heights, however, are not the best proxies to reflect vulnerabilities immediately

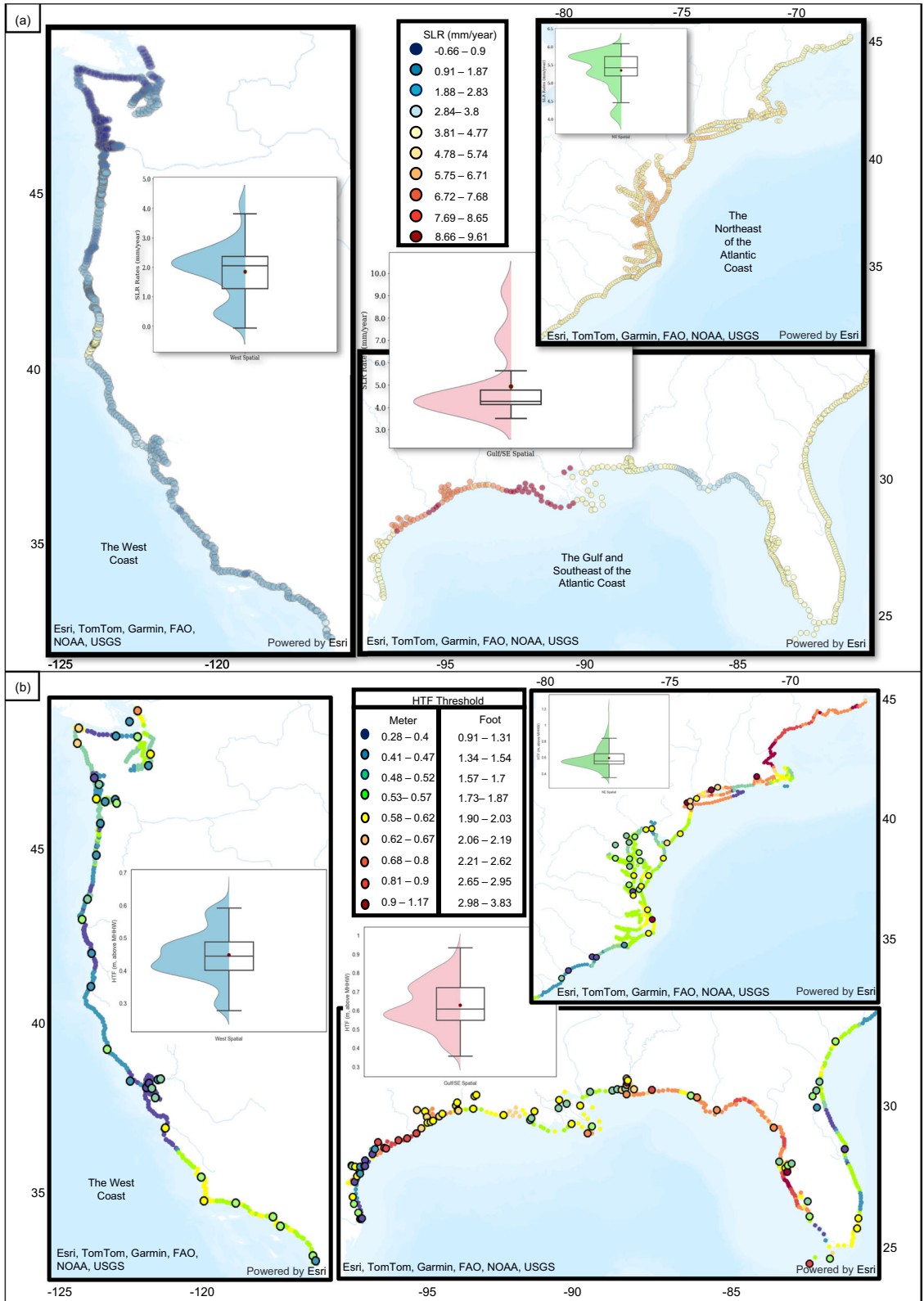

**Fig. 5 | Spatially distributed estimates of HTF thresholds and SLR rates. a** SLR rates; **b** HTF thresholds and NOAA official thresholds in larger points, the box plots show the median, 0.25, and 0.75 quantiles, maximum, and minimum of each dataset.

alongshore. The established NOAA thresholds are specifically applicable to the respective local areas where they were measured, as they solely consider the extent of HTF-induced impact in the absence of additional features during the measurement process. Thus, the impact-depth inconsistencies in NOAA thresholds pose a serious

challenge in providing useful spatially distributed information on HTF patterns/trends in places like southwest Florida, New Jersey, Texas, and near San Francisco Bay.

A better HTF thresholding system is beneficial for communication with policymakers and those tasked with coastal resources/risk

management, especially at ungauged coastal basins. Accurate flood thresholding is crucial for effective risk management and insurance industry and can provide numerous benefits for individuals, insurers, and policymakers alike. One of the main advantages of utilizing accurate flood thresholds is that insurance systems can better assess risk and organize their coverage accordingly. By understanding the maximum possible loss based on different flood mitigation strategies, insurers can more effectively provide coverage and manage risk, ultimately reducing costs for both themselves and their policyholders. For homeowners and small business owners, accurate flood thresholding can be especially valuable in planning and obtaining efficient insurance coverage. This is because these individuals are often the ones most affected by HTF, which may not be deductible through typical packages offered by the insurance companies. By having insights into the probability of HTF in their local environment, they can better prepare for potential flood-related damages and ensure that they have the appropriate insurance coverage to mitigate any losses. Governments can also benefit from accurate flood thresholding, as it enables them to allocate funds to address the cumulative impacts of HTF, especially to ensure continued infrastructure functionality. Without access to local HTF thresholds, monitoring the costs related to these events would be impossible, and so the justification of investment in measures that address cumulative minor impacts would be especially difficult.

Quantification of HTF is not only beneficial for the present condition but also for the projected flood risk management. Accurate flood thresholding can be used in new construction or redevelopment projects to reduce future HTF by implementing different strategies such as nature-based, hybrid, or gray solutions. This approach allows developers to create resilient plans that can withstand the impact of projected flooding under SLR. Generally, accurate flood thresholding can help make informed decisions regarding land-use planning and zoning regulations. Knowing the HTF threshold makes it easier to identify high-risk areas and implement measures to reduce the impact of flooding on people and infrastructure. This, in turn, can help in minimizing the potential economic losses associated with flood events. Additionally, local/regional information about HTF can help increase public awareness about the cumulative chronic impacts of SLR, which in turn can aid in the development of effective mitigation strategies.

A better thresholding development has also implications for climate impacts reports and resilience guidelines. The focus of such documents (i.e., IPCC reports) has been mainly on extreme coastal flooding trends in a warming world. The HTFs, on the other hand, although less impactful at the incident level when compared to extreme events, e.g., those with a 1% annual exceedance probability, repeatedly interrupt the traffic and yield in business closure among other health, infrastructure, and economic impacts. Thus, chronic impacts of HTF, when accumulated over time, pose a serious policy challenge that requires enhanced monitoring systems to enable risk-informed decision-making. Therefore, it is of paramount importance that reports and guidelines incorporate information and analysis of both infrequent severe events and the more frequent, less damaging events. By doing so, decision-makers and governments can be better equipped to navigate the complexities of flood risk, ensuring the formulation of effective policies and measures that address the diverse range of flood events and their impacts.

Providing spatially distributed information on SLR rates and HTF thresholds is challenging. In most cases, the input data for the ML algorithm proposed here to predict SLR rates is available at coarse resolution, so interpolation between the available points is inevitable. Dynamically downscaled input information can greatly improve the accuracy of these predictions. Also, the tradeoff between the number of input features to obtain reliable outputs is subject to further investigation. Here, based on physical principles, we found the aforementioned features relevant; however, to what extent all those features are necessary for an accurate estimation requires a detailed analysis based on comprehensive dynamical modeling.

## Methods

In Supplementary Fig. S2, we provide a flowchart illustrating the methodology employed in this study. As depicted in this figure, our approach begins with the initial step of implementing clustering on the input data pertaining to the HTF thresholding system. Subsequently, distinct Random Forest (RF) algorithms[48,49] are developed for each cluster, specifically addressing SLR rates. Following this, the application of RF algorithms for each cluster is extended to the HTF threshold model. The steps are explained in detail in the following sections. It is worth noting that each set of input data for the RF algorithms is documented with corresponding footnotes in Supplementary Table S1, which represents a comprehensive explanation of the characteristics of these input data for the reference and clarity.

### Machine learning algorithm

Different hyperparameters can be used to develop a RF regressor, the most important of which are the number of estimators, and maximum depth. These hyperparameters indicate the number of trees in the forest, and the maximum depth of the trees on which they should be expanded, respectively. All these hyperparameters were used in the development of RF algorithms in this study. Moreover, bootstrapping is used to build each tree. In the context of RF modeling, the term bootstrap refers to the practice of training each tree within the ensemble on a distinct subset of the available observations, as opposed to utilizing the entire dataset for training[50,51]. The choice of hyperparameters is critical while optimizing the model, since they can enhance the algorithm's capacity for learning and predictability[40]. Various methodologies exist for the determination of optimal hyperparameters pertaining to a given dataset, one of which is the Randomized Search technique. This approach, known for its computational efficiency, involves the exploration of user-defined combinations of hyperparameter values to select the most suitable configuration. In this research, this method is implemented to provide the hyperparameters required for the algorithms to better understand the underlying patterns within the dataset.

### Input features

Various processes contributing to the relative SLR rates are expected to yield distinct regional patterns along the CONUS coastlines. For example, while the mean sea level is expected to be 0.12–0.26 m higher in the northern West coast by the mid-21st century, the western Gulf coast is expected to experience three times more considerable rise in the relative mean sea level, i.e., 0.51–0.79 m[12]. To estimate the relative SLR, reanalysis records of global ocean heat[52], ocean circulation[53], salinity[54], sea level pressure, surface pressure, and sea surface temperature[55] were obtained at 10 km resolution from different sources and considered as input features to the RF algorithm. The vertical land motion[56] was another input feature expected to contribute to the relative SLR.

Furthermore, relative SLR, GTR, the continental shelf slope, and the coastal elevation were considered as effective variables on the HTF thresholding system to be used in the regression process. The SLR helps in assessing how HTF threshold exceedances have changed over time. The continental shelf slope is specifically essential, given its contribution to the modulation of coastal wave propagation[30]. Coastal elevation serves as a fundamental determinant in influencing the vulnerability and resilience of coastal regions, directly impacting the extent and severity of inundation during flood events. Choosing it as a key variable emphasizes its importance in understanding and dealing with how factors like topobathy, climate changes, and the coastal landscape's ability to handle and adapt to flooding challenges interact. Another aspect that merited consideration is the flood defenses, a critical parameter given its important role in flood scenarios. However,

the unavailability of easily accessible public data led to the decision not to incorporate this component into the analysis. Moreover, latitude and longitude were considered among the input variables for both processes as they are indicators of the geographical features. Supplementary Table S1 lists the various sources of data used for training and validation. These input features are selected based on the findings from previous studies[1,5,12,20,22,57]. To split the dataset into training and validation subsets, we assumed that the temporal patterns exhibited by the input features remained constant over the time frame. Consequently, the complete time series was joined into a singular entity, and individual values were assigned to each data point. Subsequently, the allocation of data points into training and validation sets was executed through a process of random sampling, resulting in an 85% allocation to the training dataset and a 15% allocation to the validation dataset.

In the training stage, ML algorithms require the input features to be accessible in every location of the desired target point. Hence, we obtained the input features for each ML algorithm at desired points. Empirical Bayesian Kriging was utilized to interpolate between the input feature data points in cases where input data had coarse resolution. The geostatistical interpolation method, Empirical Bayesian Kriging (EBK), offers an automated approach to address the challenging aspects of constructing a reliable kriging model. EBK streamlines the parameter estimation process by employing subsetting and simulations to compute the necessary parameters. By automating these key steps, EBK alleviates the burden of manual parameter calculation, enhancing the efficiency and accuracy of the kriging modeling process[58]. Next, Extract Multi Values to Point in ArcGIS Pro was used to provide the features' values at the available target points; the results of this step were used in the well-trained and validated RF algorithms to generate spatially distributed SLR rates and HTF thresholds every 10 km.

### Target variables

To train the RF algorithm, we utilized the SLR rates from Kopp et al.[22], which provided 145 SLR rates for gauges along the CONUS coastlines. They considered different climate scenarios and percentiles to deliver the SLR rates. Here we have considered the 50th percentile of the estimated SLR for the year 2020 under RCP 4.5 in the study of Kopp et al.[22] as the baseline for present-day. However, we analyzed other scenarios and presented the results in the supplementary information. Moreover, to capture the spatial variability of HTF thresholds, we developed RF regression algorithms while considering available thresholds from different sources. For validation purpose of our HTF estimation model, besides using the original records of HTF threshold provided by Sweet et al. 5, we have obtained recently estimated thresholds in one hundred additional gauges along the CONUS coastline; the information of which will soon be made publicly accessible.

During the development of algorithms for determining SLR rates and HTF thresholds, it became evident that there was an insufficient quantity of target (output) variables available. Consequently, we adopted a data augmentation technique referred to as the buffer zone method to increase the available target variables. In this approach, a circular buffer with a radius of 2.5 km was created around each existing target point. Within this buffer zone, two to three additional points were generated, and these newly generated points were assigned the same value as the original target variable. This method effectively expanded the dataset of target variables for our modeling efforts. This method was employed based on the authors' assertion that coastal characteristics exhibit minimal variation within a radius of 2.5 km from a given point.

### Clustering

HTF thresholding system was influenced by several variables, such as SLR rates, GTR, continental shelf slope, coastal elevation, latitude, and

longitude, each possessing distinct characteristics in different regions. Therefore, using only one regressor was not enough to capture the variations in input features and understand the complex patterns in the target variables. Consequently, clustering techniques were implemented to split the input features into distinct classes. Each of these classes would then be associated with its own dedicated RF regressor, equipped to effectively capture and assimilate the pertinent patterns. In this context, the K-means[35] clustering method was utilized. The K-means clustering algorithm creates clusters by considering the mean value of objects within each cluster. To achieve the optimal clustering outcome, it necessitates multiple iterations, involving various initial selections of cluster centers, for a given number of clusters[59]. A critical step in K-means clustering is detecting the optimal number of clusters, which should be adequately large to divide the datasets with different characteristics into different clusters and should be small enough to cover a sufficient number of samples in each cluster for further ML development. The next step involves the development of the RF algorithms on individual clusters for each distinct process.

### Validation

To validate the performance of the proposed RF regression algorithms in predicting unseen values and test the algorithm's prediction power, a portion of 15% of the data was kept out of the training process for both the SLR rates and HTF threshold models. The algorithm was then fed using these unseen values to evaluate its performance in predicting values not seen during training. This process was repeated 100 times, and the average evaluation metrics were used to determine how well the algorithm predicted the samples it had never seen.

### Data availability

All data used in this study is publicly accessible. Please refer to Supplementary Table S1 in the Supplementary Information file to see the full list of links to the repositories used in this study. Also, the final products of this project are accessible at https://shorturl.at/cptP1.

### Code availability

The codes, input data, and output data of the ML algorithms are deposited on our GitHub page (https://github.com/CHL-UA).

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

## Acknowledgements

This work was financially supported by USACE award no. A20-0545-001 awarded to HMor and HMof, and by the NOAA award to the Cooperative Institute for Research to Operations in Hydrology (CIROH) through the NOAA Cooperative Agreement with The University of Alabama (NA22NWS4320003) awarded to HMof. The statements and conclusions are those of the authors and do not necessary reflect the views of funding agencies.

## Author contributions

S.M. and H.Mof. conceptualized the study and designed the whole framework. S.M. collected the data, and implemented the analysis. H.Mof. supervised the work. D.F.M. helped with the model setup. W.S. provided the unpublished HTF thresholds and helped with technical adjustments and discussions. S.M. wrote the first draft of the manuscript. H.Mof., H.Mor., D.F.M. and W.S. provided comments and edited the manuscript. HMof and HMor secured funding.

## Competing interests

The authors declare no competing interests.
