## [Peer Review File · Nature Communications]

Establishing flood thresholds for sea level rise impact communicationREVIEWER COMMENTS

Reviewer #1 (Remarks to the Author):

The paper developed a machine learning model to estimate SLR rates and HTF thresholds at a relatively fine spatial resolution along the United States coastlines. Then, the authors validated the HFT thresholding system against officially reported rates and thresholds at gauges not used in the ML training phase. This work is new in the field of SLR but the paper needs more clarification and edits:

Abstract:

- In lines 17 and 21, the authors used the abbreviation "HFT" instead of "HTF".
- The sentences are long and poorly written.
- The authors should briefly add the finding from the study.

Introduction:

- The first sentence of paragraph starting in line 44 is ambiguous.
- The authors should describe that they use clustering and RF algorithm.
- The authors should mention why they use clustering and regression algorithms, i.e., what is the purpose of each model?
- The authors should clearly mention the input and output of the RF model need to be stated clearly. It is hard to get such information from the manuscript.

Methodology:

- The chosen RF hyperparameters should be stated in the methods section.
- The authors should describe the RF and clustering techniques in more details in the methods section.
- The authors mention that they used 10% of the data for testing. The authors should mention at what time interval are these estimates. Because the training data set should be trained using previous/historical years, while the test should be future dates. For example, the training data set should be used for the years up to 2020, while the testing data set should include the results from 2021 to 2023.

Results, discussion, and conclusion:

- Why do you need to do clustering? Isn't it a regression model to estimate the SLR and HTF values?
- The "discussion and conclusion" section is more like an introduction than a discussion section. Only the last paragraph is the one that discusses the results. However, it needs to be expanded.

Reviewer #2 (Remarks to the Author):

Review of "A good flood thresholding is crucial for effective sea level rise impact communication" by Mahmoudi et al.

Reviewer: Sean Vitousek

The paper uses machine learning methods (clustering + random forests (RF)) to predict rates of relative sea-level rise (SLR) and high tide flooding (HTF) thresholds across the contiguous US (CONUS) coastline. The method is trained / compared to data from Sweet et al., (2018) database of HTF (or 'minor' flood) thresholds, which reveals that minor flooding thresholds across CONUS are closely approximated with an elevation corresponding to 0.5 m above the local mean higher high water (MHHW) elevation (plus a small contribution related to the local tide range). I find the Sweet et al. (2018) relationship to be remarkably useful, but, of course, having a more precise, site-specific estimate of the flooding thresholds would be even better. So, I think the paper is very well motivated.

Unfortunately, I think many aspects of the paper need to be improved before the paper is ready for publication, in my opinion. I argue that rather major revisions are needed before the paper is further considered for publication in Nature Communications.

I wish that I could be a bit more positive about the paper, since I strongly agree that flood thresholds (and their spatial variability) are not well understood and are often overlooked. And enhancing the understanding gained in Sweet et al. would be a worthy endeavor, in my opinion.

A few critical areas for improvement are listed below:

1. A better discussion on varying coastal elevation is needed. You mention in line 62-63 that "relying solely on tidal records may not be sufficient" to identifying flooding thresholds, which is certainly true. And in my opinion, coastal elevation is simply the most critical factor. And, I might recommend explicitly mentioning how the elevation of coastal infrastructure (among other things) is a critical factor in flooding, in addition to coastal water level.

Although coastal elevation is a critical factor in flooding, it is quite remarkable how Sweet et al., (2018) found that the quasi-static threshold of about MHHW+0.5 is a skillful predictor of minor flooding thresholds. And I think the simplicity of this threshold rule is a testament of humanities propensity to build as close to the ocean as possible.

Throughout the literature (and in many of my own papers), flooding threshold proxies are identified by investigating 1-year, 20-year, 50-year, etc. water-level events and making the assumption that most coastal infrastructure is designed to be situated above these thresholds (but just barely). The Sweet approach of making observations of flood thresholds is obviously far superior, but with significant effort. Using ML to better understand the spatial variability of these thresholds is a very good idea, but it is rather poorly motivated/discussed, in my opinion.

2. A lot of the methodological content / equations should probably be relegated to the Methods section at the end of the manuscript, to be more consistent with Nature formatting. I also think that only the equation $HTF_MHHW = 0.04 \times GTR + 0.5$ (i.e., Eq 3) is worth keeping. All of the other math/equations are trivial and unnecessary, IMO.

3. If one were to predict a HTF threshold at a particular location, then they could use one of ~3 approaches: 1) use the Sweet et al., relationship (i.e., $MHHW + 0.5 + 0.04 \times GTR$), 2) find the nearest neighboring observation of the minor flood threshold from Sweet et al., and use that, or 3) the developed ML method. In Figure 4, you make comparisons (on average across the regions) between the Sweet et al., linear relationship, observations, and the ML method, and the ML seems to closely resemble the observations. But why not just use the nearest observation for determining the threshold? Is there something about the current ML method that can better estimate the HTF threshold better than simply a nearest neighbor interpolation of the observations? If so, then this would provide a very good motivation for the current method, I think.

4. As far as I'm concerned, the ML method is just a very fancy interpolation method, which is certainly not a bad thing (and please feel free correct me if you feel that this is not the right opinion to hold). In essence, you're trying to learn the underlying patterns from the Sweet minor flood threshold observations and then use them to predict the thresholds at locations where observations are lacking. Most of the model inputs have to do with oceanographic variables, i.e., relative SLR, tide range, shelf slope, as well as "reanalysis records of global ocean heat, ocean circulation, salinity, sea level pressure, surface pressure, and sea surface temperature were obtained at 10 km resolution from different sources and considered as input features to our RF algorithm.", which I honestly don't think would add much (since Sweet's $MHHW + 0.5 + 0.04 \times GTR$ seems to fit most of the observations well). You do include geographic variables like latitude and longitude elevation, but, I think including terrestrial variables like elevation (even in an average sense), might be much more appropriate to

include. There is likely to be a relationship between elevation and shelf slope, but it might be better to use elevation itself rather than elevation slope. Is there a reason why elevation is not included as an input? Certainly that is more important than salinity to determine the flood threshold ...

5. The discussion needs to be refined quite a bit, I think. The first four paragraphs of the discussion seem to only to provide further motivation for why HTF thresholds are important, but this has already been well established in the introduction. Instead, I think the discussion conclusions should try to put this work more in the context of other works rather than try to provide additional motivation.

Also, the discussion on where the ML-derived HTF thresholds strongly differ from the Sweet et al. linear relationship is rather lacking. I feel there are huge differences between the ML-derived HTF thresholds and the Sweet MHHW+0.5 threshold in places. And this is particularly evident in the Gulf where the tide ranges are rather low but the HTF thresholds are high. The authors make a comment about Hurricane barriers as being the cause, but I think more discussion is needed. Perhaps there are a few minor flood threshold observations in the Sweet database that might be considered "outliers" (when compared to the simple linear regression formula), and they control these high threshold behavior in the Gulf. Hence, it might be nice to plot the Sweet observations on Figure 5, as well, but with a different marker type.

Specific comments:

Title: "A good flood thresholding is crucial for effective sea level rise impact communication" The title does very little to motivate what the paper is about, in my opinion. That is, the manuscript is not about 'communication'... the paper is about a model to estimate HTF thresholds.

Line 12-14: "Monitoring evolving patterns ..."
Needs a bit of revision, in my opinion.

Line 25-26: "While at
26 the incident level HTF might not seem ..." might need some revision

Line 59-60: "Remote Sensing data" ... more specifics needed on the source of the remote sensing data ... satellite altimetry?

Line 72-73: Might want to be more specific about what you mean by "nearby"...

For example, this line might be combined with the following line "Consequently, they have no better option than making significant assumptions and using information available at the nearest tide gauge that might be located hundreds of kilometers away, with its own accuracy challenges"

Line 83: Greater Tide Range (GTR). You might want to mention somewhere that $GTR = MHHW - MLLW$, as it becomes more apparent/useful in the subsequent analysis

Figure 1 - A lot of the discussion in the introduction of the math/equations and figure 1 seem a bit unnecessary. Yes, the difference between datums (i.e., MLLW and MHHW are noticeable, and yes MHHW is probably better to use for flooding), but conversion between datums is a bit too trivial to discuss in such depth, IMO.

Figure 2 - What variables are you clustering on (latitude and longitude)? If so then you might just consider separating the West, Gulf, and East Coasts a priori. It is interesting that the clustering method basically does is automatically, though.

Line 235: "... proposed by ref 5" might consider the last name, e.g., Sweet et al. (ref 5) instead of just "ref", but it is probably up to the journal convention.

Figure 4a. I might recommend two different panels as it is very hard to with so many points to tell the difference between the ML method and the linear regression method (LR).

Figure 4b. I might recommend making a dashed line at 0.5 m since that is the baseline HTF flood elevation based on Sweet. Everything from the new method should be compared to that, I think.

ML method: Withholding only 10% of the data seems low. Can this choice be further motivated?

Responses to Reviewers' comments on "A good flood thresholding is crucial for effective sea level rise impact communication" by Mahmoudi et al.; Article reference: NCOMMS-23-30690 to Nature Communications.

We really appreciate the thoughtful comments and constructive suggestions that significantly helped improve the quality of this work. For clarity, we have included the original reviewers' comments in blue text and our point-by-point response in black text. All line numbers in this response file refer to the revised submission currently under review.

Reviewer #1

The paper developed a machine learning model to estimate SLR rates and HTF thresholds at a relatively fine spatial resolution along the United States coastlines. Then, the authors validated the HFT thresholding system is against officially reported rates and thresholds at gauges not used in the ML training phase. This work is new in the field of SLR but the paper needs more clarification and edits.

Thanks for the supportive feedback, and also helpful suggestions that significantly improved the quality of work. Please, see the lines below on how your comments are fully implemented in this revision.

Abstract:

1. In lines 17 and 21, the authors used the abbreviation "HFT" instead of "HTF".

Sorry for the typo. It's fixed in the current submission.

2. The sentences are long and poorly written.

We have re-written portions of the abstract with long sentences, and we believe the revised version of abstract has an improved flow and readability. The revised version of abstract reads:

"Sea level rise (SLR) affects coastal flood regimes and poses a serious challenge to flood risk management, particularly in ungauged or unmonitored coasts. To track SLR and its impacts, appropriate methods are needed to effectively monitor SLR trends at local to regional scales and the associated flooding thresholds. To address this, we propose a high tide flood (HTF) thresholding system that leverages machine learning (ML) techniques to estimate SLR rates and HTF thresholds at a relatively fine spatial resolution along the United States' coastlines. The developed algorithms are trained and validated against officially reported values at monitoring gauges operated by National Oceanic and Atmospheric Administration (NOAA). The proposed system complements conventional linear- and point-based estimations of SLR rates and HTF thresholds and can help estimate these values at ungauged stretches of the coast. The products can raise community awareness about SLR impacts by documenting the chronic signal of HTF and providing useful information for adaptation planning and management. The results of the proposed system show promising skills in estimating the NOAA's thresholds using ML with an average KGE of 0.77, especially at the three-fourth of the United States' coastal points without a tide gauge within a 10 km radius. The findings encourage further application of ML in achieving spatially distributed thresholds."

3. The authors should briefly add the finding from the study.

Some of the findings were added to the abstract (lines 22-25) as:

"The results of the proposed system show promising skills in estimating the NOAA's thresholds using ML with an average KGE of 0.77, especially at the three-fourth of the United States' coastal points

without a tide gauge within a 10 km radius. The findings encourage further application of ML in achieving spatially distributed thresholds.”

Introduction:

4. The first sentence of paragraph starting in line 44 is ambiguous.

This sentence has been revised for further clarity (lines 49-51). The revised version reads:

“Significant progress has been made in projecting SLR using hybrid approaches that combine process-based modeling with statistical methods. These approaches enable us to quantify uncertainties and likelihoods of various SLR projections under different climate change scenarios.”

5. The authors should describe that they use clustering and RF algorithm.

We added the use of clustering and RF algorithm in the new version of Introduction.

The use of clustering is added to the lines 140-144 as: *“Moreover, the observed variability in the components that impact the HTF thresholding system has made it necessary to apply clustering techniques. These techniques help partition effective variables into distinct categories classified by shared characteristics. The clustering approach has played a pivotal role in generating more accurate and refined outcomes pertinent to HTF thresholds.”*

The use of RF algorithms is added to the lines 144-146 as: *“The Random Forest (RF) regression model was selected as an appropriate ML model because of its ability to identify underlying patterns and predict the values of HTF thresholds at ungauged basins.”*

6. The authors should mention why they use clustering and regression algorithms, i.e., what is the purpose of each model?

While regression algorithms are useful for estimating target variables (i.e., SLR rates and HTF thresholds), the input features for each target variable contain variability across the case study (CONUS coastlines). In recognition of this inherent variability and the spatial complexities of the domain, it became evident that relying solely on a single, overarching regression model would not yield sufficiently reliable results. To address this issue and enhance the accuracy of predictions, clustering techniques were introduced to facilitate the identification of an optimal number of clusters, specifically concerning the influential features impacting HTF thresholds. These techniques help partition the influential variables into distinct categories characterized by shared characteristics. By doing so, the clustering approach contributes significantly to refining and enhancing the precision of the outcomes.

The use of regression methods is added to the Introduction (lines 137-140) as: *“ML algorithms can effectively process and analyze numerous input features simultaneously. In the last two decades, the use of ML methods for extracting characteristics, trends, or rules from datasets has been rapidly increasing for data-intensive hydrological modeling problems^{33,34}.”*

The reason to use clustering is also added to the Introduction (lines 140-144): *“Moreover, the observed variability in the components that impact the HTF thresholding system has made it necessary to apply clustering techniques. These techniques help partition effective variables into distinct categories classified by shared characteristics. The clustering approach has played a pivotal role in generating more accurate and refined outcomes pertinent to HTF thresholds.”*

7. The authors should clearly mention the input and output of the RF model need to be stated clearly. It is hard to get such information from the manuscript.

We agree that the RF model and its inputs/outputs should have been explained with more details. Thus, sections 4.2 and 4.3 were added to the Methodology and the input and output of the RF model are clearly stated under the “**Input Features**” and “**Target Variables**” sections.

The input variables were added to the lines 459-481 as: “*Various processes contributing to the relative SLR rates are expected to yield distinct regional patterns along the CONUS coastlines. For example, while the mean sea level is expected to be 0.12-0.26 m higher in the northern West coast by the mid-21st century, the western Gulf coast is expected to experience three times more significant rise in the relative mean sea level, i.e., 0.51-0.79 m12. To estimate the relative SLR, reanalysis records of global ocean heat52, ocean circulation53, salinity54, sea level pressure, surface pressure, and sea surface temperature55 were obtained at 10 km resolution from different sources and considered as input features to the RF algorithm. The vertical land motion56 was another input feature expected to contribute to the relative SLR.*

Furthermore, relative SLR, GTR, the continental shelf slope, and the coastal elevation were considered as effective variables on the HTF thresholding system to be used in the regression process. The SLR helps in assessing how HTF threshold exceedances have changed over time. The continental shelf slope is specifically essential, given its contribution to the modulation of coastal wave propagation30. Coastal elevation serves as a fundamental determinant in influencing the vulnerability and resilience of coastal regions, directly impacting the extent and severity of inundation during flood events. Choosing it as a key variable emphasizes its importance in understanding and dealing with how factors like topobathy, climate changes, and the coastal landscape's ability to handle and adapt to flooding challenges interact. Another aspect that merited consideration is the flood defenses, a critical parameter given its significant role in flood scenarios. However, the unavailability of easily accessible public data led to the decision not to incorporate this component into the analysis. Moreover, latitude and longitude were considered among the input variables for both processes as they are indicators of the geographical features. Table S1 lists the various sources of data used for training and validation. These input features are selected based on the findings from previous studies^{1,5,12,20,22,57}.”

The target variables were appended in lines 500-509 as: “*To train the RF algorithm, we utilized the SLR rates from Kopp et al.22, which provided 145 SLR rates for gauges along the CONUS coastlines. They considered different climate scenarios and percentiles to deliver the SLR rates. Here we have considered the 50th percentile of the estimated SLR for the year 2020 under RCP 4.5 in the study of Kopp et al.22 as the baseline for present-day. However, we analyzed other scenarios and presented the results in the supplementary information. Moreover, to capture the spatial variability of HTF thresholds, we developed RF regression algorithms while considering available thresholds from different sources. For validation purpose of our HTF estimation model, besides using the original records of HTF threshold provided by Sweet et al.5, we have obtained recently estimated thresholds in one hundred additional gauges along the CONUS coastline; the information of which will soon be made publicly accessible.*”

Methodology:

8. The chosen RF hyperparameters should be stated in the methods section.

Thanks for the comment. The pertinent information is now added in lines 444-450 “*Different hyperparameters can be used to develop a RF regressor, the most important of which are the number of estimators, and maximum depth. These hyperparameters indicate the number of trees in the forest, and the maximum depth of the trees on which they should be expanded, respectively. All these hyperparameters*

were used in the development of RF algorithms in this study. Moreover, bootstrapping is used to build each tree. In the context of RF modeling, the term bootstrap refers to the practice of training each tree within the ensemble on a distinct subset of the available observations, as opposed to utilizing the entire dataset for training”

We also provided the values of hyperparameters for SLR rates and HTF thresholds in Tables S2 and S5, respectively.

9. The authors should describe the RF and clustering techniques in more details in the methods section.

The Methods section is now revised (significantly expanded) to include more detailed information of RF algorithms for regression and K-means algorithm for clustering. The revised section (lines 434-443) reads as:

“In this study, we use the RF algorithm⁴⁸, which is a supervised learning technique that generates an ensemble of decision trees and aggregates their predictions to predict continuous numerical values. The superior accuracy, robustness, and scalability of RF regression algorithms have made them a popular choice over other ML algorithms, particularly when dealing with spatiotemporal datasets that contain numerous input variables or when the relationships between variables are intricate. Decision trees employ a tree-like structure resembling a flowchart to illustrate predictions derived from sequential feature-based divisions. Beginning with a root node, they culminate in decisions determined by leaf nodes. RF is implemented to learn the underlying pattern at existing SLR rates and HTF thresholds and acquire them at ungauged locations along the coastlines of the CONUS in Atlantic Ocean, Pacific Ocean, and the Gulf of Mexico⁴⁹⁻⁵¹. ”

The clustering method was added to the **Clustering** subsection of the Methodology section lines 520-534 *“HTF thresholding system was influenced by several variables, such as SLR rates, GTR, continental shelf slope, coastal elevation, latitude, and longitude, each possessing distinct characteristics in different regions. Therefore, using only one regressor was not enough to capture the variations in input features and understand the complex patterns in the target variables. Consequently, clustering techniques were implemented to split the input features into distinct classes. Each of these classes would then be associated with its own dedicated RF regressor, equipped to effectively capture and assimilate the pertinent patterns. In this context, the K-means³⁵ clustering method was utilized. The K-means clustering algorithm creates clusters by considering the mean value of objects within each cluster. To achieve the optimal clustering outcome, it necessitates multiple iterations, involving various initial selections of cluster centers, for a given number of clusters⁵⁹. A critical step in K-means clustering is detecting the optimal number of clusters, which should be adequately large to divide the datasets with different characteristics into different clusters and should be small enough to cover a sufficient number of samples in each cluster for further ML development. The next step involves the development of the RF algorithms on individual clusters for each distinct process. ”*

10. The authors mention that they used 10% of the data for testing. The authors should mention at what time interval are these estimates. Because the training data set should be trained using previous/historical years, while the test should be future dates. For example, the training data set should be used for the years up to 2020, while the testing data set should include the results from 2021 to 2023.

In contrary with the process-based costal hydrodynamics modeling with a longitudinal calibration and validation process (i.e., the timespan of training and validation sets are different), here we are employing a

cross-sectional training and validation approach at which the patterns of the input feature in a period of time are assumed to be time-invariant. Hence, we decided to treat the entire sample size as a whole and take a single value for each point. Then for training and validation process, the random sampling was done on the entire points to obtain training and validation datasets. This information was added to the manuscript in lines 481-486:

“To split the dataset into training and validation subsets, we assumed that the temporal patterns exhibited by the input features remained constant over the time frame. Consequently, the complete time series was joined into a singular entity, and individual values were assigned to each data point. Subsequently, the allocation of data points into training and validation sets was executed through a process of random sampling, resulting in an 85% allocation to the training dataset and a 15% allocation to the validation dataset.”

Results, discussion, and conclusion:

11. Why do you need to do clustering? Isn't it a regression model to estimate the SLR and HTF values?

Given the inherent variability in the input features affecting HTF thresholding system across the CONUS coastlines, it became evident that relying solely on a single, comprehensive regression model would not produce the desired level of reliability in the results. To overcome this challenge and improve the accuracy of predictions, we introduced clustering techniques to determine the most appropriate number of clusters, particularly with regard to the influential features affecting HTF thresholds. These techniques aid in categorizing the influential variables into distinct groups defined by common characteristics. This clustering approach, in turn, plays a pivotal role in refining and enhancing the precision of the outcomes.

The explanation on the reason to use clustering technique was added to the Methodology in lines 520-534 *“HTF thresholding system was influenced by several variables, such as SLR rates, GTR, continental shelf slope, coastal elevation, latitude, and longitude, each possessing distinct characteristics in different regions. Therefore, using only one regressor was not enough to capture the variations in input features and understand the complex patterns in the target variables. Consequently, clustering techniques were implemented to split the input features into distinct classes. Each of these classes would then be associated with its own dedicated RF regressor, equipped to effectively capture and assimilate the pertinent patterns. In this context, the K-means³⁵ clustering method was utilized. The K-means clustering algorithm creates clusters by considering the mean value of objects within each cluster. To achieve the optimal clustering outcome, it necessitates multiple iterations, involving various initial selections of cluster centers, for a given number of clusters⁵⁹. A critical step in K-means clustering is detecting the optimal number of clusters, which should be adequately large to divide the datasets with different characteristics into different clusters and should be small enough to cover a sufficient number of samples in each cluster for further ML development. The next step involves the development of the RF algorithms on individual clusters for each distinct process.”*

12. The “discussion and conclusion” section is more like an introduction than a discussion section. Only the last paragraph is the one that discusses the results. However, it needs to be expanded.

The discussion section is now extended to cover the results (lines 313-417). However, we kept the benefits of a better thresholding system to be a proof that our study and results can act as a communication tool between different disciplinaries.

The new version of discussion is updated based on your comment and other reviewer's.

First, we discussed the common approaches to achieving the local thresholds and the inconsistency and challenge with each of the approaches *“This study aimed at improving the existing coastal HTF thresholding system. The two commonly used approaches are 1) based on specific impacts at each location, similar to the NWS method, which does not consistently represent the same severity of impact in their thresholding approach across places of interest, and 2) considering influential features on the thresholding system. The latter approach aims at establishing a unified method applicable across all regions, which mitigates the necessity of establishing local monitoring stations to record flood impacts for threshold determination. Therefore, transitioning and enhancing the thresholding system from a locally impact-based approach to a regionally feature-based methodology is imperative in advancing flood monitoring and management practices. To pursue this approach, Sweet et al.⁵ developed a linear regression based on one variable, which was found to be limited, constrained, and non-generalizable, especially in regions with higher GTR values, such as the North Atlantic Coast. Moreover, focusing on a single variable in the HTF thresholding system may lead to negligence of other crucial factors.”*

Then a brief explanation on the methodology was given *“The proposed methodology in this study represents a significant advancement towards a global strategy that leverages ML algorithms to provide spatially distributed information efficiently. Among the influential components of the HTF thresholding system, SLR rates played a pivotal role, given their significant impact on coastal flooding events. However, the localized availability of SLR rates presented a challenge, demanding a deeper investigation into this critical variable. To address this challenge and derive local SLR rates and HTF thresholds, we employed a ML algorithm, which offers the advantage of including multiple effective variables.”*

The results were then elaborated on. First, we discussed the SLR rates’ results *“The literature^{12,44} suggested that SLR rates in Louisiana and Texas are 2-4 times higher than global average (3.1 ± 0.3 mm/yr^{16,19-21}), which confirms our findings (Figure 5b). These elevated rates are attributed to multiple factors, including shallow sediment compaction, hydrological changes, and fluid extraction⁴⁴⁻⁴⁶. On the other hand, SLR rates are lower than the global average rates in the West coast especially because of isostatic adjustments⁴⁷.”*

Then we compared our results with the study of Sweet et al. in lines 338-345 *“The outcome of this study differs from the results of the linear regressions (LR) method introduced by Sweet et al.⁵, who attempted to normalize and constrain the NOAA official thresholds in a range. The LR method demonstrates a relatively acceptable performance primarily in areas where the HTF threshold hovers within approximately 0.5 meters above the MHHW. Nevertheless, it is noteworthy that our ML algorithm exhibits a remarkable capacity to generate reasonably accurate results across a more extensive spectrum of observed HTF threshold ranges. The natural differences in HTF thresholds across regions highlight limitations in current methods that mainly use univariate linear regression and suggest a need to shift to more advanced, nonlinear approaches.”*

Afterward, we explained the results of HTF threshold by introducing the significant factor contributing to the HTF thresholds in different regions *“Based on Table S7, one significant factor affecting HTF thresholds in the Northeast coast is the GTR. Specifically, it is observed that GTR values are higher in the states of Massachusetts, New Hampshire, and Maine, which contributes to the elevated HTF thresholds observed therein. Moreover, based on Table S7, SLR stands as a more influential component on the HTF thresholds than GTR in the Gulf and Southeast coasts. Hence, higher rates of SLR could generally lead to higher HTF thresholds, such as the east Texas, that exhibits high SLR rates and HTF thresholds. While Louisiana faces high SLR rates, the lack of comprehensive flood mitigation strategies has led to lower HTF thresholds in this region, partially owed to the impact-based thresholding system*

set by NOAA. This complex situation makes Louisiana more prone to HTF effects and underscores how the complex interplay of environmental factors and human actions shapes coastal vulnerabilities. ”

Next, we defined the discrepancies in HTF thresholds due to the inconsistency with the NOAA official thresholds “Some disparities are evident between HTF estimates in specific localities and their adjacent parts (Figure 5b), which results in spatial heterogeneity near flood prone metros. For example, a flood protection provided by hurricane seawalls in regions from Corpus Christi to Galveston Bay in Texas yields in higher HTF thresholds in this region, compared with other parts of Texas and the coast of Louisiana. Furthermore, the flood mitigation strategies employed in New York made the HTF thresholds be higher than their surroundings. These heights, however, are not the best proxies to reflect vulnerabilities immediately alongshore. In fact, the established NOAA thresholds are specifically applicable to the respective local areas where they were measured, as they solely consider the extent of HTF-induced impact in the absence of additional features during the measurement process. Thus, the impact-depth inconsistencies in NOAA thresholds pose a serious challenge in providing useful spatially distributed information on HTF patterns/trends in places like southwest Florida, New Jersey, Texas, and near San Francisco Bay. ”

Finally, the advantages of a better thresholding system and its implications across diverse contexts were defined (this section was slightly modified as a wrap up to our study) in lines 369-417: “A better HTF thresholding system is beneficial for communication with policymakers and those tasked with coastal resources/risk management, especially at ungauged coastal basins. Accurate flood thresholding is crucial for effective risk management and insurance industry and can provide numerous benefits for individuals, insurers, and policymakers alike. One of the main advantages of utilizing accurate flood thresholds is that insurance systems can better assess risk and organize their coverage accordingly. By understanding the maximum possible loss based on different flood mitigation strategies, insurers can more effectively provide coverage and manage risk, ultimately reducing costs for both themselves and their policyholders. For homeowners and small business owners, accurate flood thresholding can be especially valuable in planning and obtaining efficient insurance coverage. This is because these individuals are often the ones most affected by HTF, which may not be deductible through typical packages offered by the insurance companies. By having insights into the probability of HTF in their local environment, they can better prepare for potential flood-related damages and ensure that they have the appropriate insurance coverage to mitigate any losses. Governments can also benefit from accurate flood thresholding, as it enables them to allocate funds to address the cumulative impacts of HTF, especially to ensure continued infrastructure functionality. Without access to local HTF thresholds, monitoring the costs related to these events would be impossible, and so the justification of investment in measures that address cumulative minor impacts would be especially difficult.

Quantification of HTF is not only beneficial for the present condition but also for the projected flood risk management. Accurate flood thresholding can be used in new construction or redevelopment projects to reduce future HTF by implementing different strategies such as nature-based, hybrid, or grey solutions. This approach allows developers to create resilient plans that can withstand the impact of projected flooding under SLR. In fact, accurate flood thresholding can help make informed decisions regarding land-use planning and zoning regulations. Knowing the HTF threshold makes it easier to identify high-risk areas and implement measures to reduce the impact of flooding on people and infrastructure. This, in turn, can help in minimizing the potential economic losses associated with flood events. Additionally, local/regional information about HTF can help increase public awareness about the cumulative chronic impacts of SLR, which in turn can aid in the development of effective mitigation strategies.

A better thresholding development has also implications for climate impacts reports and resilience guidelines. The focus of such documents (i.e., IPCC reports) has been mainly on extreme coastal flooding trends in a warming world. The HTFs, on the other hand, although less impactful at the incident level when compared to extreme events, e.g., those with a 1% annual exceedance probability, repeatedly interrupt the traffic and yield in business closure among other health, infrastructure, and economic impacts. Thus, chronic impacts of HTF, when accumulated over time, pose a serious policy challenge that requires enhanced monitoring systems to enable risk-informed decision-making. Therefore, it is of paramount importance that reports and guidelines incorporate information and analysis of both infrequent severe events and the more frequent, less damaging events. By doing so, decision-makers and governments can be better equipped to navigate the complexities of flood risk, ensuring the formulation of effective policies and measures that address the diverse range of flood events and their impacts.

Providing spatially distributed information on SLR rates and HTF thresholds is though challenging. In most cases, the input data for the ML algorithm proposed here to predict SLR rates is available at coarse resolution, so interpolation between the available points is inevitable. Dynamically downscaled input information can significantly improve the accuracy of these predictions. Also, the tradeoff between the number of input features to obtain reliable outputs is subject to further investigation. Here, based on physical principles, we found the aforementioned features relevant; however, to what extent all those features are necessary for an accurate estimation requires a detailed analysis based on comprehensive dynamical modeling.”

Reviewer #2

The paper uses machine learning methods (clustering + random forests (RF)) to predict rates of relative sea-level rise (SLR) and high tide flooding (HTF) thresholds across the contiguous US (CONUS) coastline. The method is trained / compared to data from Sweet et al., (2018) database of HTF (or ‘minor’ flood) thresholds, which reveals that minor flooding thresholds across CONUS are closely approximated with an elevation corresponding to 0.5 m above the local mean higher high water (MHHW) elevation (plus a small contribution related to the local tide range). I find the Sweet et al. (2018) relationship to be remarkably useful, but, of course, having a more precise, site-specific estimate of the flooding thresholds would be even better. So, I think the paper is very well motivated.

Thanks for the summary and supportive feedback.

Unfortunately, I think many aspects of the paper need to be improved before the paper is ready for publication, in my opinion. I argue that rather major revisions are needed before the paper is further considered for publication in Nature Communications.

We agree that the manuscript can be further improved based on your constructive suggestions. So, in the following lines we show how your helpful hints and comments are fully implemented in the revisions.

1. A better discussion on varying coastal elevation is needed. You mention in line 62-63 that "relying solely on tidal records may not be sufficient" to identifying flooding thresholds, which is certainly true. And in my opinion, coastal elevation is simply the most critical factor. And, I might recommend explicitly mentioning how the elevation of coastal infrastructure (among other things) is a critical factor in flooding, in addition to coastal water level.

In the previous version of our analysis, we considered the variation of coastal elevation in the form of the continental shelf slope, which covered both the elevation (from GEBCO bathymetry data) and the slope. The literature suggested that *“The continental shelf slope is specifically essential, given its contribution to the modulation of coastal wave propagation.”* That is the reason we considered the continental shelf slope.

In this submission, we have explicitly considered the coastal elevation (from GEBCO bathymetry data) alongside the continental shelf slope and carried out the RF regressor. The choice of considering coastal elevation was added to the **Input Features** sub-section of the Methodology section lines 471-475 as: *“Coastal elevation serves as a fundamental determinant in influencing the vulnerability and resilience of coastal regions, directly impacting the extent and severity of inundation during flood events. Choosing it as a key variable emphasizes its importance in understanding and dealing with how factors like topobathy, climate changes, and the coastal landscape's ability to handle and adapt to flooding challenges interact.”*

However, the evaluation metrics showed worsened results in comparison with the time when coastal elevation and continental shelf slope were omitted from the analysis. Hence, we described both scenarios' results and decided to go without coastal elevation and continental shelf slope. Moreover, a new figure was added to the supplementary information (Figure S2), which shows the percentile ranking of the difference between the official and ML-predicted thresholds with and without considering continental shelf slope and coastal elevation. As shown in this figure, the percentile ranking within ± 0.05 m difference (an acceptable difference) drops when considering coastal elevation and continental shelf slope. The discussion on this matter was added to the sub-section of the HTF algorithm in the Result section in lines 257-267 as: *“During this stage of analysis, a necessary refinement was performed by removing two components, including coastal elevation and continental shelf slope, which were found to have relatively minimal impacts on the thresholding system. This strategic elimination of less influential factors led to improvements in the model performance and a decrease of computational time to train the RF regressor. As depicted in Figure S3, a compelling representation of the percentile ranking was developed based on the disparities between the officially established thresholds by NOAA and the values predicted through the RF algorithms. In this context, disparities falling within the narrow range of ± 0.05 m were assumed to be negligible. However, disparities exceeding the defined threshold were regarded as substantial deviations warranting attention. Figure S3 demonstrates that the percentile ranking associated with significant disparities shows an increase when the coastal elevation and continental shelf slope were omitted from the input features.”*

Moreover, “infrastructure (flood defenses) elevation” was another alternative to be considered in the analysis; however, the information regarding the infrastructure elevation is not easily and publicly accessible in a large domain like the CONUS. This explanation was added to the **Input Features** sub-section of the Methodology section lines 475-478 as: *“Another aspect that merited consideration is the flood defenses, a critical parameter given its significant role in flood scenarios. However, the unavailability of easily accessible public data led to the decision not to incorporate this component into the analysis.”*

Although coastal elevation is a critical factor in flooding, it is quite remarkable how Sweet et al., (2018) found that the quasi-static threshold of about MHHW+0.5 is a skillful predictor of minor flooding thresholds. And I think the simplicity of this threshold rule is a testament of humanities propensity to build as close to the ocean as possible.

Throughout the literature (and in many of my own papers), flooding threshold proxies are identified by investigating 1-year, 20-year, 50-year, etc. water-level events and making the assumption that most coastal infrastructure is designed to be situated above these thresholds (but just barely). The Sweet approach of making observations of flood thresholds is obviously far superior, but with significant effort. Using ML to better understand the spatial variability of these thresholds is a very good idea, but it is rather poorly motivated/discussed, in my opinion.

We agree that a better justification for more sophisticated approaches (i.e., ML) compared with the linear regression method proposed by Sweet et al., is necessary to clearly elaborate on needs that motivated our study. In this submission, we have further elaborated on the motivation behind this study in the Introduction (lines 125-135) as:

“Moreover, this means a linear regression solely based on one independent variable (here GTR) generally fails to capture the stochastic nature of spatial variability in HTF threshold above the local high tide datum (i.e., MHHW). This approach, if generalized as done in previous studies^{12,13,28,31,32}, could yield in significant error in estimated flood frequency. Therefore, the variability in HTF thresholds necessitates the incorporation of multiple features, and a simple linear regression method proves insufficient to unravel the complex patterns inherent in these thresholds.

These facts motivated us to develop an approach that provides flood thresholds above the MHHW datum (i.e., HTF_MHHW) based on multiple physically relevant variables, which can inform decision-makers and emergency managers about the patterns and efficiency of mitigation plans coping with nuisance flooding.”

We also added explanations on the use of ML algorithms in the Methodology section in lines 428-443 as: *“ML algorithms are capable of fitting nonlinear models to sample data, also called training data, and further making predictions or classifications using new or unseen sample data that is referred to as validation data. To develop the ML algorithms, first, feature transformation was implemented on the input features. Feature transformation refers to the process of converting input features into a new representation that is more suitable for training ML algorithms. Extracting more useful and relevant information from the original input features can improve the accuracy and effectiveness of ML models. In this study, we use the RF algorithm⁴⁸, which is a supervised learning technique that generates an ensemble of decision trees and aggregates their predictions to predict continuous numerical values. The superior accuracy, robustness, and scalability of RF regression algorithms have made them a popular choice over other ML algorithms, particularly when dealing with spatiotemporal datasets that contain numerous input variables or when the relationships between variables are intricate. Decision trees employ a tree-like structure resembling a flowchart to illustrate predictions derived from sequential feature-based divisions. Beginning with a root node, they culminate in decisions determined by leaf nodes. The RF is implemented to learn the underlying patterns at existing SLR rates and HTF thresholds and acquire them at ungauged locations along the coastlines of the CONUS⁴⁹⁻⁵¹.”*

We added the use of ML algorithms and the reason in the Discussion section in lines 331-332 as: *“To address this challenge and derive local SLR rates and HTF thresholds, we employed a ML algorithm, which offers the advantage of including multiple effective variables.”*

2. A lot of the methodological content / equations should probably be relegated to the Methods section at the end of the manuscript, to be more consistent with Nature formatting. I also think that only the equation $HTF_{MHHW} = 0.04 \times GTR + 0.5$ (i.e., Eq 3) is worth keeping. All of the other math/equations are trivial and unnecessary, IMO.

We appreciate the intention of respected reviewer in improving the flow of manuscript. And, we do agree that unnecessary equations, if included in the manuscript, may adversely affect the readability of manuscript. While we see some equation rearrangements helpful (i.e. in response to your 12th comment, we moved Eq 2 ($HTF_{MLLW} = GTR + HTF_{MHHW}$) to the point we first introduce GTR (lines 102-103) and some other equations to the Methods section, we believe Equations 1 and 3 are still worth keeping as they best serve the Introduction to help audience unfamiliar with the basics of coastal tides fully grasp the idea behind this work and how it is different than the previous studies. We believe such methodological background

complemented by the revised version of Figure 1, helps most effectively highlight the research gap and contribution of this study.

3. If one were to predict a HTF threshold at a particular location, then they could use one of ~3 approaches: 1) use the Sweet et al., relationship (i.e., $MHHW+0.5+0.04*GTR$), 2) find the nearest neighboring observation of the minor flood threshold from Sweet et al., and use that, or 3) the developed ML method. In Figure 4, you make comparisons (on average across the regions) between the Sweet et al., linear relationship, observations, and the ML method, and the ML seems to closely resemble the observations. But why not just use the nearest observation for determining the threshold? Is there something about the current ML method that can better estimate the HTF threshold better than simply a nearest neighbor interpolation of the observations? If so, then this would provide a very good motivation for the current method, I think.

The ML approach learns the underlying patterns in the input features (effective variables on HTF threshold) and predict the target variables (HTF thresholds) based on those patterns. If we are to use nearest neighbor values, we would neglect the patterns and assume the desired place and the nearest official threshold location share the same characteristics (effective variables) which is not the case in many regions. Also, the distance between the desired point and the nearest official threshold location might be more than a hundred kilometer which make it illogical to use that threshold when we already have a better option like ML approach.

Hence, we implemented an analysis that divides the US coastlines into 10 km intervals (we assume within 10 km the coastal characteristics share insignificant difference) and finds the nearest tide gauge to each location. Figure S1 (posted here below for your convenience) shows the distribution of the distance between the nearest tide gauge and each HTF estimations location. As shown in this figure, 75% of the locations do not have access to a tide gauge within their 10 km proximity, which disables us from accessing to reliable data.

This information was added to the lines 78-90 of Introduction to further clarify on the motives behind this project: *“If the US coastlines were segmented into 10 km intervals, accommodating variations in coastal characteristics, 75% of the coastal communities in the United States lack access to tide gauges within a 10 km radius of their respective locations (Figure S1). This observation underscores the limited coverage of and accessibility to tide gauge infrastructure along the U.S. coastlines, thereby hindering emergency managers and stakeholders from accessing reliable local information including HTF thresholds and SLR rates. Consequently, they have no better option than making significant assumptions and using information available at the nearest tide gauge that might be located more than a hundred kilometers away, with its own accuracy and uncertainty challenges²⁸. The utilization of nearest tide gauge values necessitates careful consideration, as it might overlook the underlying patterns of the influential features on the HTF thresholding system. This approach presumes that the target ungauged point and the nearest official threshold location inherently share consistent characteristics in terms of the influential variables.”*

4. As far as I'm concerned, the ML method is just a very fancy interpolation method, which is certainly not a bad thing (and please feel free correct me if you feel that this is not the right opinion to hold). In essence, you're trying to learn the underlying patterns from the Sweet minor flood threshold observations and then use them to predict the thresholds at locations where observations are lacking. Most of the model inputs have to do with oceanographic variables, i.e., relative SLR, tide range, shelf slope, as well as “reanalysis records of global ocean heat, ocean circulation, salinity, sea level pressure, surface pressure, and sea surface temperature were obtained at 10 km resolution from different sources and considered as input features to our RF algorithm.”, which I honestly don't think would add much (since Sweet's $MHHW+0.5+0.04*GTR$ seems to fit most of the observations well). You do include geographic variables like latitude and longitude elevation, but, I think including terrestrial variables like elevation (even in an average sense), might be much more appropriate to include. There is likely to be a relationship between elevation and shelf slope, but it might be better to use elevation itself rather than elevation slope. Is there a reason why elevation is not included as an input? Certainly that is more important than salinity to determine the flood threshold ...

While both ML and interpolation involve predicting values between known data points, there are important distinctions: 1. ML models, such as neural networks or random forests, can capture complex and nonlinear relationships among the input features, whereas traditional interpolation methods, like linear or spline interpolation, assume simpler relationships between the target variable and input features. 2. ML aims at generalizing patterns learned from data to make predictions on new, unseen data. Interpolation, on the other hand, is typically used to estimate values at points within the range of existing data. In summary, while both interpolation and ML involve making predictions between data points, ML is a more versatile and powerful approach that can capture complex relationships and generalize patterns beyond the training data, so theoretically ML supposed to be a better choice for this research, compared with simple interpolation methods.

Here, we are trying to predict two processes. Sea level rise and HTF thresholds. The oceanographic data (global ocean heat, ocean circulation, salinity, sea level pressure, surface pressure, and sea surface temperature) are used to predict SLR rates at fine spatial resolution; whereas SLR rates, GTR, and shelf slope are used to predict HTF thresholds. We understand our original write up was not clearly distinguishing between these processes and their influential variable. Hence, we added a new sub section as **Input Features** to the Methodology (lines 459-479) as:

“Various processes contributing to the relative SLR rates are expected to yield distinct regional patterns along the CONUS coastlines. For example, while the mean sea level is expected to be 0.12-0.26 m higher

in the Northwest Coasts of the US by the mid-21st century, the western Gulf Coast is expected to experience three times more significant rise in mean sea level, i.e., 0.51-0.79 m¹². To estimate the relative SLR, reanalysis records of global ocean heat⁵², ocean circulation⁵³Error! Reference source not found., salinity⁵⁴, sea level pressure, surface pressure, and sea surface temperature⁵⁵ were obtained at 10 km resolution from different sources and considered as input features to the RF algorithm. The vertical land motion⁵⁶ was another input feature expected to contribute to the relative SLR.

Furthermore, relative SLR, GTR, the continental shelf slope, and coastal elevation were considered as effective variables on the HTF thresholding system to be used in the regression process. The SLR helps in assessing how HTF threshold exceedances have changed over time. The continental shelf slope is specifically essential, given its contribution to the modulation of coastal wave propagation³⁰. Coastal elevation serves as a fundamental determinant in influencing the vulnerability and resilience of coastal regions, directly impacting the extent and severity of inundation during flood events. Choosing it as a key variable emphasizes its importance in understanding and dealing with how factors like topobathy, climate changes, and the coastal landscape's ability to handle and adapt to flooding challenges interact. Another aspect that merited consideration is the flood defenses, a critical parameter given its significant role in flood scenarios. However, the unavailability of easily accessible public data led to the decision not to incorporate this component into the analysis. Moreover, latitude and longitude were considered among the input variables for both processes as they are indicators of the geographical features.”

We also provided a schematic of our work that demonstrates the target variables with their associated input features in Figure S2. This information was added to the manuscript in lines 419-426 as: *“In Figure S2, we provide a flowchart illustrating the methodology employed in this study. As depicted in this figure, our approach begins with the initial step of implementing clustering on the input data pertaining to the HTF thresholding system. Subsequently, distinct RF algorithms are developed for each cluster, specifically addressing SLR rates. Following this, the application of RF algorithms for each cluster is extended to the HTF threshold model. The steps are explained in detail in the following sections. It is worth noting that each set of input data for the RF algorithms is documented with corresponding footnotes in Table S1, which represents a comprehensive explanation of the characteristics of these input data for reference and clarity.”*

Also, thanks for the helpful suggestion. We considered the coastal elevation in the updated version of our manuscript and analysis. We have this variable among the list of input features for our ML algorithms (lines 471-475) and included in new results (see Tables S6a and S8b).

However, upon examination, it was noted that the evaluation metrics exhibited a decline in performance when compared to the results obtained without the inclusion of coastal elevation and continental shelf slope in the analysis. Consequently, both sets of results were documented and, after careful consideration, the decision was made to proceed without incorporating coastal elevation and continental shelf slope. To supplement these findings, a new addition was made to the supplementary information in the form of Figure S3. This figure provides a visual representation of the percentile ranking, illustrating the difference between the official and ML-predicted thresholds with and without the consideration of continental shelf slope and coastal elevation. Notably, the comparison between panels a and b in Figure S3 highlights that the percentile ranking, specifically within the range of ± 0.05 m difference (which is considered an acceptable variance), experiences a decrease when coastal elevation and continental shelf slope are taken into account. A corresponding discussion on this matter has been integrated into the subsection focusing on the HTF algorithm within the Results section, commencing at the lines 257-267: *“During this stage of analysis, a necessary refinement was performed by removing two components, including coastal elevation and continental shelf slope, which were found to have relatively minimal impacts on the thresholding system. This strategic elimination of less influential factors led to improvements in the model performance and a*

decrease of computational time to train the RF regressor. As depicted in Figure S3, a compelling representation of the percentile ranking was developed based on the disparities between the officially established thresholds by NOAA and the values predicted through the RF algorithms. In this context, disparities falling within the narrow range of ± 0.05 m were assumed to be negligible. However, disparities exceeding the defined threshold were regarded as substantial deviations warranting attention. Figure S3 demonstrates that the percentile ranking associated with significant disparities shows an increase when the coastal elevation and continental shelf slope were omitted from the input features.”

5. The discussion needs to be refined quite a bit, I think. The first four paragraphs of the discussion seem to only to provide further motivation for why HTF thresholds are important, but this has already been well established in the introduction. Instead, I think the discussion conclusions should try to put this work more in the context of other works rather than try to provide additional motivation.

Thank you for your comment. In the revised version of our manuscript, in the Discussions section, first we have explained the different approaches available to achieve HTF thresholds. The approaches consist of the method employed by WFO in the National Weather Service (NWS) and the one by Sweet et al. Then, we stated the challenges with these approaches. This information was added to the lines 312-323 as: “*This study aimed at improving the existing coastal HTF thresholding system. The two commonly used approaches are 1) based on specific impacts at a given location, and 2) considering influential features on the thresholding system. While the former suffers from the consistency of severity of impacts between various locations, the latter approach aims at establishing a unified method applicable across all regions, therefore mitigating the necessity of establishing local monitoring stations to record flood impacts for threshold determination. Transitioning and enhancing the thresholding system from an impact-based local approach to a regionally feature-based methodology is imperative in advancing flood monitoring and management practices. To pursue this approach, Sweet et al.⁵ developed a linear regression based on one variable, which was found to be constrained, and non-generalizable, especially in regions with higher GTR values, such as the Northeast coast of the United States. Moreover, focusing on a single variable in the HTF thresholding system may lead to negligence of other crucial factors.*”

Considering all the challenges, we briefly explained our methodology and why using ML algorithms is beneficial in this case in lines 326-332: “*The proposed methodology in this study represents a significant advancement towards a global strategy that leverages ML algorithms to provide spatially distributed information efficiently. Among the influential components of the HTF thresholding system, SLR rates played a pivotal role, given their significant impact on coastal flooding events. However, the localized availability of SLR rates presented a challenge, demanding a deeper investigation into this critical variable. To address this challenge and derive local SLR rates and HTF thresholds, we employed a ML algorithm, which offers the advantage of including multiple effective variables.*”

The results were then elaborated on. First, we discussed the SLR rates’ results “*The literature^{12,44} suggested that SLR rates in Louisiana and Texas are 2-4 times higher than global average (3.1 ± 0.3 mm/yr^{16,19-21}), which confirms our findings (Figure 5b). These elevated rates are attributed to multiple factors, including shallow sediment compaction, hydrological changes, and fluid extraction^{Error! Reference source not found.44-46}. On the other hand, SLR rates are lower than the global average rates in the West Coast especially because of isostatic adjustments⁴⁷.*”

We compared our results with the study of Sweet et al. in lines 338-345: “*The outcome of this study differs from the results of the LR method introduced by Sweet et al.⁵, who attempted to normalize and constrain the NOAA official thresholds in a range. The LR method demonstrates a relatively acceptable performance*

primarily in areas where the HTF threshold hovers within approximately 0.5 meters above the MHHW. Nevertheless, it is noteworthy that our ML algorithm exhibits a remarkable capacity to generate reasonably accurate results across a more extensive spectrum of observed HTF threshold ranges. The natural differences in HTF thresholds across regions highlight limitations in current methods that mainly use univariate linear regression and suggest a need to shift to more advanced, nonlinear approaches.”

Afterward, we explained the results of HTF threshold by introducing the significant factor contributing to the HTF thresholds in different regions “*Based on Table S7, one significant factor affecting HTF thresholds in the North Atlantic is the GTR. Specifically, it is observed that GTR values are higher in the states of Massachusetts, New Hampshire, and Maine, which contributes to the elevated HTF thresholds observed therein. Moreover, based on Table S7, SLR stands as a more influential component on the HTF thresholds than GTR in the Gulf and South of the Atlantic. Hence, higher rates of SLR should lead to higher HTF thresholds, such as the east Texas, that exhibits high SLR rates and HTF thresholds. While Louisiana faces high SLR rates, the lack of comprehensive flood mitigation strategies has led to lower HTF thresholds in this region, because of the impact-based thresholding system set by NOAA. This complex situation makes Louisiana more prone to HTF effects and underscores how the complex interplay of environmental factors and human actions shapes coastal vulnerabilities.”*

Next, we defined the discrepancies in HTF thresholds due to the inconsistency with the NOAA official thresholds “*Some disparities are evident between HTF estimates in specific localities and their adjacent parts (Figure 5b), which results in spatial heterogeneity near flood prone metros. For example, a flood protection provided by hurricane seawalls in regions from Corpus Christi to Galveston Bay in Texas yields in higher HTF thresholds in this region, compared with other parts of Texas and the coast of Louisiana. Furthermore, the flood mitigation strategies employed in New York made the HTF thresholds be higher than their surroundings. These heights, however, are not the best proxies to reflect vulnerabilities immediately alongshore. In fact, the established WFO thresholds are specifically applicable to the respective local areas where they were measured, as they solely consider the extent of HTF-induced impact in the absence of additional features during the measurement process. Thus, the impact-depth inconsistencies in WFO thresholds pose a serious challenge in providing useful spatially distributed information on HTF patterns/trends in places like southwest Florida, New Jersey, Texas, and near San Francisco Bay. Thus, to overcome this shortcoming, spatially recognizing vulnerabilities is key for planning in the face of SLR.”*

Finally, the advantages of a better thresholding system and its implications across diverse contexts were defined (this section was slightly modified as a wrap up to our study and proof that our findings can be used to a better communication between scientists and policymakers) in lines 369-417: “*A better HTF thresholding system is beneficial for communication with policymakers and those tasked with coastal resources/risk management, especially at ungauged coastal basins. Accurate flood thresholding is crucial for effective risk management and insurance industry and can provide numerous benefits for individuals, insurers, and policymakers alike. One of the main advantages of utilizing accurate flood thresholds is that insurance systems can better assess risk and organize their coverage accordingly. By understanding the maximum possible loss based on different flood mitigation strategies, insurers can more effectively provide coverage and manage risk, ultimately reducing costs for both themselves and their policyholders. For homeowners and small business owners, accurate flood thresholding can be especially valuable in planning and obtaining efficient insurance coverage. This is because these individuals are often the ones most affected by HTF, which may not be deductible through typical packages offered by the insurance companies. By having insights into the probability of HTF in their local environment, they can better prepare for potential flood-related damages and ensure that they have the appropriate insurance coverage*

to mitigate any losses. Governments can also benefit from accurate flood thresholding, as it enables them to allocate funds to address the cumulative impacts of HTF, especially to ensure continued infrastructure functionality. Without access to local HTF thresholds, monitoring the costs related to these events would be impossible, and so the justification of investment in measures that address cumulative minor impacts would be especially difficult.

Quantification of HTF is not only beneficial for the present condition but also for the projected flood risk management. Accurate flood thresholding can be used in new construction or redevelopment projects to reduce future HTF by implementing different strategies such as nature-based, hybrid, or grey solutions. This approach allows developers to create resilient plans that can withstand the impact of projected flooding under SLR. In fact, accurate flood thresholding can help make informed decisions regarding land-use planning and zoning regulations. Knowing the HTF threshold makes it easier to identify high-risk areas and implement measures to reduce the impact of flooding on people and infrastructure. This, in turn, can help in minimizing the potential economic losses associated with flood events. Additionally, local/regional information about HTF can help increase public awareness about the cumulative chronic impacts of SLR, which in turn can aid in the development of effective mitigation strategies.

A better thresholding development has also implications for climate impacts reports and resilience guidelines. The focus of such documents (i.e., IPCC reports) has been mainly on extreme coastal flooding trends in a warming world. The HTFs, on the other hand, although less impactful at the incident level when compared to extreme events, e.g., those with a 1% annual exceedance probability, repeatedly interrupt the traffic and yield in business closure among other health, infrastructure, and economic impacts. Thus, chronic impacts of HTF, when accumulated over time, pose a serious policy challenge that requires enhanced monitoring systems to enable risk-informed decision-making. Therefore, it is of paramount importance that reports and guidelines incorporate information and analysis of both infrequent severe events and the more frequent, less damaging events. By doing so, decision-makers and governments can be better equipped to navigate the complexities of flood risk, ensuring the formulation of effective policies and measures that address the diverse range of flood events and their impacts.

Providing spatially distributed information on SLR rates and HTF thresholds is though challenging. In most cases, the input data for the ML algorithm proposed here to predict SLR rates is available at coarse resolution, so interpolation between the available points is inevitable. Dynamically downscaled input information can significantly improve the accuracy of these predictions. Also, the tradeoff between the number of input features to obtain reliable outputs is subject to further investigation. Here, based on physical principles, we found the aforementioned features relevant; however, to what extent all those features are necessary for an accurate estimation requires a detailed analysis based on comprehensive dynamical modeling.”

6. Also, the discussion on where the ML-derived HTF thresholds strongly differ from the Sweet et al. linear relationship is rather lacking. I feel there are huge differences between the ML-derived HTF thresholds and the Sweet MHHW+0.5 threshold in places. And this is particularly evident in the Gulf where the tide ranges are rather low but the HTF thresholds are high. The authors make a comment about Hurricane barriers as being the cause, but I think more discussion is needed. Perhaps there are a few minor flood threshold observations in the Sweet database that might be considered “outliers” (when compared to the simple linear regression formula), and they control these high threshold behavior in the Gulf. Hence, it might be nice to plot the Sweet observations on Figure 5, as well, but with a different marker type.

We compared the work of Sweet et al. with our findings in the discussion in lines 338-345 as *“The outcome of this study differs from the results of the LR method introduced by Sweet et al.⁵, who attempted to normalize and constrain the NOAA official thresholds in a range. The LR method demonstrates a relatively acceptable performance primarily in areas where the HTF threshold hovers within approximately 0.5 meters above the MHHW. Nevertheless, it is noteworthy that our ML algorithm exhibits a remarkable capacity to generate reasonably accurate results across a more extensive spectrum of observed HTF threshold ranges. The natural differences in HTF thresholds across regions highlight limitations in current methods that mainly use univariate linear regression and suggest a need to shift to more advanced, nonlinear approaches.”*

We also stated why HTF thresholds might show discrepancies to their vicinities in lines 346-355 *“Based on Table S7, one significant factor affecting HTF thresholds in the North Atlantic is the GTR. Specifically, it is observed that GTR values are higher in the states of Massachusetts, New Hampshire, and Maine, which contributes to the elevated HTF thresholds observed therein. Moreover, based on Table S7, SLR stands as a more influential component on the HTF thresholds than GTR in the Gulf and South of the Atlantic. Hence, higher rates of SLR should lead to higher HTF thresholds, such as the east Texas, that exhibits high SLR rates and HTF thresholds. While Louisiana faces high SLR rates, the lack of comprehensive flood mitigation strategies has led to lower HTF thresholds in this region, because of the impact-based thresholding system set by NOAA. This complex situation makes Louisiana more prone to HTF effects and underscores how the complex interplay of environmental factors and human actions shapes coastal vulnerabilities.”*

There is a possibility for existence of outliers in the NOAA official thresholds. The reason is due to the fact that WFO uses impacts to develop the thresholds and does not consider any variables affecting the thresholding system. Hence, in places with established flood defenses having higher HTF thresholds are logical. However, considering the flood defenses as an input feature to the ML algorithms requires public and accessible data about the defenses, which are not available in many states.

The observation of Sweet et al. is demonstrated in Figure 5b now in larger bullet points. Figure 5b shows that ML does its best to even predict the outliers.

7. Title: *“A good flood thresholding is crucial for effective sea level rise impact communication”* The title does very little to motivate what the paper is about, in my opinion. That is, the manuscript is not about ‘communication’... the paper is about a model to estimate HTF thresholds.

Although this manuscript is about ML algorithms to estimate HTF thresholds, the objective of this estimation is to be able to better monitor and communicate the sea level rise impacts on high-frequency low-impact flooding regime in coastal region. Without the proper quantification of HTF thresholds, we will not be able to deliver such information (rates and patterns of potential influence); hence, we cannot plan towards proper adaptation and mitigation strategies. We have further clarified this in the Introduction and Discussion section as:

lines 132-135 *“These facts motivated us to develop an approach that provides flood thresholds above the MHHW datum (i.e., HTF_{MHHW}) based on multiple physically relevant variables, which can inform decision-makers and emergency managers about the mitigation plans coping with nuisance flooding.”*

Lines 151-157 *“This study presents the first continental-scale estimates of high spatial resolution data on relative SLR rates and HTF thresholds, which are intended to facilitate the communication of SLR risk to vulnerable coastal communities, particularly in locations that lack gauge measurements. Our findings, utilizing a well-trained and validated ML algorithm, provide reliable information at a spatial resolution (10 km along the U.S. coastline) significantly finer than the previously available datasets^{Error! Reference source}”*

not found.^{5,12}. This information plays a pivotal role in effective risk-informed decision-making and communication of trending flood hazards associated with SLR to the at-risk coastal communities.”

Lines 369-417 “A better HTF thresholding system is beneficial for communication with policymakers and those tasked with coastal resources/risk management, especially at ungauged coastal basins. Accurate flood thresholding is crucial for effective risk management and insurance industry and can provide numerous benefits for individuals, insurers, and policymakers alike. One of the main advantages of utilizing accurate flood thresholds is that insurance systems can better assess risk and organize their coverage accordingly. By understanding the maximum possible loss based on different flood mitigation strategies, insurers can more effectively provide coverage and manage risk, ultimately reducing costs for both themselves and their policyholders. For homeowners and small business owners, accurate flood thresholding can be especially valuable in planning and obtaining efficient insurance coverage. This is because these individuals are often the ones most affected by HTF, which may not be deductible through typical packages offered by the insurance companies. By having insights into the probability of HTF in their local environment, they can better prepare for potential flood-related damages and ensure that they have the appropriate insurance coverage to mitigate any losses. Governments can also benefit from accurate flood thresholding, as it enables them to allocate funds to address the cumulative impacts of HTF, especially to ensure continued infrastructure functionality. Without access to local HTF thresholds, monitoring the costs related to these events would be impossible, and so the justification of investment in measures that address cumulative minor impacts would be especially difficult.

Quantification of HTF is not only beneficial for the present condition but also for the projected flood risk management. Accurate flood thresholding can be used in new construction or redevelopment projects to reduce future HTF by implementing different strategies such as nature-based, hybrid, or grey solutions. This approach allows developers to create resilient plans that can withstand the impact of projected flooding under SLR. In fact, accurate flood thresholding can help make informed decisions regarding land-use planning and zoning regulations. Knowing the HTF threshold makes it easier to identify high-risk areas and implement measures to reduce the impact of flooding on people and infrastructure. This, in turn, can help in minimizing the potential economic losses associated with flood events. Additionally, local/regional information about HTF can help increase public awareness about the cumulative chronic impacts of SLR, which in turn can aid in the development of effective mitigation strategies.

A better thresholding development has also implications for climate impacts reports and resilience guidelines. The focus of such documents (i.e., IPCC reports) has been mainly on extreme coastal flooding trends in a warming world. The HTFs, on the other hand, although less impactful at the incident level when compared to extreme events, e.g., those with a 1% annual exceedance probability, repeatedly interrupt the traffic and yield in business closure among other health, infrastructure, and economic impacts. Thus, chronic impacts of HTF, when accumulated over time, pose a serious policy challenge that requires enhanced monitoring systems to enable risk-informed decision-making. Therefore, it is of paramount importance that reports and guidelines incorporate information and analysis of both infrequent severe events and the more frequent, less damaging events. By doing so, decision-makers and governments can be better equipped to navigate the complexities of flood risk, ensuring the formulation of effective policies and measures that address the diverse range of flood events and their impacts.

Providing spatially distributed information on SLR rates and HTF thresholds is though challenging. In most cases, the input data for the ML algorithm proposed here to predict SLR rates is available at coarse resolution, so interpolation between the available points is inevitable. Dynamically downscaled input information can significantly improve the accuracy of these predictions. Also, the tradeoff between the number of input features to obtain reliable outputs is subject to further investigation. Here, based on

physical principles, we found the aforementioned features relevant; however, to what extent all those features are necessary for an accurate estimation requires a detailed analysis based on comprehensive dynamical modeling.”

8. Line 12-14: “Monitoring evolving patterns of SLR and their impacts on coastal areas demand robust methods that can effectively regionalize information of SLR trends and associated flood thresholds.” Needs a bit of revision, in my opinion.

Changed to *“To track SLR and its impacts, appropriate methods are needed to effectively monitor SLR trends at local to regional scales and the associated flooding threshold.”*

9. Line 25-26: “ While at the incident level HTF might not seem to be very significant, monitoring its evolution over time and its associated impacts on the resilience of coastal communities is of paramount importance. ” might need some revision.

Changed to *“At the incident level, HTF may not pose significant damages to the coastal communities. However, monitoring its evolution over time and assessing its associated impacts on the resilience of coastal communities is crucial.”* in lines 29-32.

10. Line 59-60: “Remote Sensing data” ... more specifics needed on the source of the remote sensing data ... satellite altimetry?

We changed it to satellite altimetry in lines 64-65.

11. Line 72-73: Might want to be more specific about what you mean by “nearby” in “In fact, along the U.S. coasts, there exist thousands of communities without a tide gauge in a nearby location.”. For example, this line might be combined with the following line “Consequently, they have no better option than making significant assumptions and using information available at the nearest tide gauge that might be located hundreds of kilometers away, with its own accuracy challenges”

We conducted an analysis segmenting the US coastlines into 10 km intervals, where we assume that coastal characteristics exhibit insignificant differences within each 10 km segment. We then identified the nearest tide gauge to each location within these segments. Figure S1 illustrates the distribution of distances between the nearest tide gauge and each location. It is evident from this figure that approximately 75% of these locations lack access to a tide gauge within their 10 km proximity, which consequently hinders our access to reliable data.

We added this information in lines 77-83 *“If the US coastlines were segmented into 10 km intervals, accommodating variations in coastal characteristics, 75% of the coastal communities in the United States lack access to tide gauges within a 10 km radius of their respective locations (Figure S1). This observation underscores the limited coverage of and accessibility to tide gauge infrastructure along the U.S. coastlines, thereby hindering emergency managers and stakeholders from accessing reliable local information including HTF thresholds and SLR rates.”*

12. Line 83: Greater Tide Range (GTR). You might want to mention somewhere that $GTR = MHHW - MLLW$, as it becomes more apparent/useful in the subsequent analysis.

This information was added the first time we introduced GTR in lines 99-103: *“They proposed a linear relationship between thresholds above mean lower low water (MLLW; the average of the lower low water height of each tidal day observed over the National Tidal Datum Epoch) and greater tidal ranges (GTR; which is the difference between MLLW and MHHW datums; $MLLW - MHHW = GTR$)^{29,30} as a means to establish a globally applicable HTF thresholding system.”*

13. Figure 1 - A lot of the discussion in the introduction of the math/equations and figure 1 seem a bit unnecessary. Yes, the difference between datums (i.e., MLLW and MHHW are noticeable, and yes MHHW is probably better to use for flooding), but conversion between datums is a bit too trivial to discuss in such depth, IMO.

The conversion between the datums is now relocated to the first time we introduced GTR in lines 102-103. Hence, we no longer have this equation in the introduction as it is insufficient.

However, we maintain Equations 1 and 3 as these equations play a pivotal role in the Introduction section, where they facilitate readers unfamiliar with the fundamental concepts of coastal tides in comprehending the essence of our work and its differentiation from prior studies.

Moreover, figure 1 shows how the technique employed by Sweet et al. can be challenging when converted to other datums. It also familiarizes the audiences with limited background and knowledge about different definitions in this specific case to better understand the underlying challenges.

However, you are right that the discussion seems a bit unnecessary. In the new version of the manuscript, we made the discussion on this matter briefer. The lines were reduced from 40 to 36. The new version of this section is as: “Sweet et al.⁵ aimed at addressing existing knowledge gaps regarding HTF thresholds. They found a lack of uniformity in representing the severity of flood impacts even in areas where thresholds were established. To address this issue, Sweet et al.⁵ introduced a normalization method that can be applied universally and establishes consistency in how different flood events are categorized and assessed. They proposed a linear relationship between thresholds above mean lower low water (MLLW; the average of the lower low water height of each tidal day observed over the National Tidal Datum Epoch) and greater tidal ranges (GTR; which is the difference between MLLW and MHHW datums; $MLLW - MHHW = GTR$)^{29,30} as a means to establish a globally applicable HTF thresholding system:

$$HTF_{MLLW} = 1.04 \times GTR + 0.5 \quad (1)$$

Their method measures the HTF above MLLW, mainly because NOAA’s Weather Forecast Office (WFO) originally sets similar thresholds based on MLLW. However, proposing the HTF threshold above MLLW (i.e. HTF_{MLLW}) would have an embedded GTR that might overshadow the variability of the actual flooding threshold above the mean higher high water (MHHW; the average of the higher high water height of each tidal day observed over the National Tidal Datum Epoch) in places with larger tidal ranges. In fact, MHHW should be the preferred and more logical vertical datum of reference when analyzing floods, because, locally, MHHW works as a proxy for the high tide that coastal communities expect the coastal water level to reach on a regular basis. Thus, a better formulation of HTF threshold above local high tide datum (HTF_{MHHW}) is:

$$HTF_{MHHW} = HTF_{MLLW} - GTR = 0.04 \times GTR + 0.5 \quad (2)$$

Figure 1. a) Comparison between the relationships of HTF thresholds with GTR when measured above MLLW vs. above MHHW; b) tidal characteristics and datums over a typical tidal cycle. Magenta dots and the left y-axis shows HTF_{MLLW} ; purple dots, and right y-axis represent HTF_{MHHW} in (a).

Figure 1a shows the results of the linear regressions above MLLW and MHHW datums between the officially reported HTF thresholds at 70 NOAA tide gauges along the U.S. coasts with their associated GTR. The magenta and purple dots and dashed lines show officially reported thresholds by NOAA and the linear regression results, above MLLW and MHHW datums, respectively, along with their goodness of fit metrics reported inside the box of the same color. The difference between estimates based on equations 1 and 2 relies on the different datum above which the threshold is measured. Indeed, an insufficient skill of equation 1 to estimate HTF_{MHHW} is evident, especially for points with larger GTR. Moreover, this means a linear regression solely based on one independent variable (here GTR) generally fails to capture the stochastic nature of spatial variability in HTF threshold above the local high tide datum (i.e., MHHW) and, if generalized as done in previous studies^{12,13,28,31,32}, would yield in significant error in estimated flood frequency. Therefore, the variability in HTF thresholds necessitates the incorporation of multiple features, and a simple linear regression method proves insufficient to unravel the complex patterns inherent in these thresholds.”

14. Figure 2 – What variables are you clustering on (latitude and longitude)? If so then you might just consider separating the West, Gulf, and East Coasts a priori. It is interesting that the clustering method basically does is automatically, though.

The clustering was made on SLR rates, GTR, land slope, latitude, and longitude (the input features for HTF thresholding system; at the locations of HTF official thresholds). We added some sentences in the

Methodology to further explain what variables were used in the clustering process (lines 520-523) “*HTF thresholding system was influenced by several variables, such as SLR rates, GTR, continental shelf slope, coastal elevation, latitude, and longitude, each possessing distinct characteristics in different regions. Therefore, using only one regressor was not enough to capture the variations in input features and understand the complex patterns in the target variables.*”

Also, we provided a flowchart in our supplementary information (Figure S2) showing the data the clustering is implemented on.

Also yes, we could come up with a more or less the same clustering priori, but we wanted to objectively delineate the clusters and interestingly it came back with such clear clustering that leaves no room for criticizing subjectivity of method.

15. Line 235: "... proposed by ref 5" might consider the last name, e.g., Sweet et al. (ref 5) instead of just "ref", but it is probably up to the journal convention.

Changed to “Sweet et al.⁵”

16. Figure 4a. I might recommend two different panels as it is very hard to with so many points to tell the difference between the ML method and the linear regression method (LR).

Thanks for the good suggestion. In the revised version of the manuscript, this has been divided into two different panels (Figures 4a and 4b).

17. Figure 4b. I might recommend making a dashed line at 0.5 m since that is the baseline HTF flood elevation based on Sweet. Everything from the new method should be compared to that, I think.

We have added the dashed line to this Figure (now Figure 4c).

18. ML method: Withholding only 10% of the data seems low. Can this choice be further motivated?

In the new round of analysis, we changed the validation portion to 15%. The entire dataset should be divided into 2 different subsets: Training and Testing (80% and 20%). Then, the training dataset should be divided into 2 classes: Training and Validation (80% and 20%), which will lead to a 15% validation dataset of the entire dataset.

This information was modified in the **Validation** sub-section of Methodology section in lines 536-541: “*To validate the performance of the proposed RF regression algorithms in predicting unseen values and test the algorithm's prediction power, a portion of 15% of the data was kept out of the training process for both the SLR rates and HTF threshold models. The algorithm was then fed using these unseen values to evaluate its performance in predicting values not seen during training. This process was repeated 100 times, and the average evaluation metrics were used to determine how well the algorithm predicted the samples it had never seen.*”

We also added the division of dataset into validation and training in the **Input Feature** of the Methodology section in lines 484-486 as: “*Subsequently, the allocation of data points into training and validation sets was executed through a process of random sampling, resulting in an 85% allocation to the training dataset and a 15% allocation to the validation dataset.*”

References

- Cai, J., Luo, J., Wang, S., Yang, S., 2018. Feature selection in machine learning: A new perspective. *Neurocomputing* 300, 70–79. <https://doi.org/10.1016/j.neucom.2017.11.077>
- Dhal, P., Azad, C., 2022. A comprehensive survey on feature selection in the various fields of machine learning. *Appl Intell* 52, 4543–4581. <https://doi.org/10.1007/s10489-021-02550-9>

REVIEWERS' COMMENTS

Reviewer #1 (Remarks to the Author):

The authors addressed the comments in a detailed way and fixed all the related concerns. The paper is ready for publication.

Reviewer #2 (Remarks to the Author):

Re-review of "A better flood thresholding is crucial for effective sea level rise impact communication"

Reviewer: Sean Vitousek

I appreciate the authors comments and revisions in response to my suggestions. It seems like the authors have made quite considerable text edits to their manuscript. I am now happy to recommend acceptance of the manuscript. I am also happy to see that the authors have included the derived flooding thresholds in an accompanying data set. This paper/dataset should be valuable for the coastal engineering community.

I only have a few trivial corrections, given below:

- Acronym KGE in the abstract probably needs a definition or rewording.
- Some reference numbers need to be formatted to have proper superscripting, but this is something that can probably be easily handled in typesetting/production.
- Line 103: $GTR = MHHW - MLLW$?